# Phosphoregulation accommodates Type III secretion and assembly of a tether of ER-*Chlamydia* inclusion membrane contact sites

**Rachel J Ende**[1†]**, Rebecca L Murray**[1†]**, Samantha K D'Spain**[1]**, Isabelle Coppens**[2]**, Isabelle Derré**[1]*****

[1]Department of Microbiology, Immunology, and Cancer Biology, University of Virginia School of Medicine, Charlottesville, United States; [2]Department of Molecular Microbiology and Immunology, Johns Hopkins School of Public Health, Baltimore, United States

**Abstract** Membrane contact sites (MCS) are crucial for nonvesicular trafficking-based interorganelle communication. Endoplasmic reticulum (ER)–organelle tethering occurs in part through the interaction of the ER resident protein VAP with FFAT motif-containing proteins. FFAT motifs are characterized by a seven amino acidic core surrounded by acid tracks. We have previously shown that the human intracellular bacterial pathogen *Chlamydia trachomatis* establishes MCS between its vacuole (the inclusion) and the ER through expression of a bacterial tether, IncV, displaying molecular mimicry of eukaryotic FFAT motif cores. Here, we show that multiple layers of host cell kinase-mediated phosphorylation events govern the assembly of the IncV–VAP tethering complex and the formation of ER-Inclusion MCS. Via a C-terminal region containing three CK2 phosphorylation motifs, IncV recruits CK2 to the inclusion leading to IncV hyperphosphorylation of the noncanonical FFAT motif core and serine-rich tracts immediately upstream of IncV FFAT motif cores. Phosphorylatable serine tracts, rather than genetically encoded acidic tracts, accommodate Type III-mediated translocation of IncV to the inclusion membrane, while achieving full mimicry of FFAT motifs. Thus, regulatory components and post-translational modifications are integral to MCS biology, and intracellular pathogens such as *C. trachomatis* have evolved complex molecular mimicry of these eukaryotic features.

**\*For correspondence:**
id8m@virginia.edu

[†]These authors contributed equally to this work

**Competing interest:** The authors declare that no competing interests exist.

## Editor's evaluation

This paper describes an interesting case of molecular mimicry where a *Chlamydia* protein needs to be phosphorylated by host kinases in order to interact with an host factor and recruit the host endoplasmic reticulum membrane around bacterial inclusions in the cell. Post-translational modifications of translocated bacterial effectors by host factors have rarely been reported. As such, the story is of interest to those studying microbial pathogenesis and also to cell biologists studying related mechanisms. One limitation of the study is that it relies on overexpression of tagged bacterial proteins. It will therefore be important to confirm these findings in the future with endogenous proteins, when adequate tools become available.

## Introduction

In naive cells, membrane contact sites (MCS) are points of contact between the membrane of two adjacent organelles (10–30 nm apart). They provide physical platforms for the nonvesicular transfer of lipids and ions, and cell signaling events important for interorganelle communication and organelle positioning and dynamics (*Prinz et al., 2020*). Since their discovery and implication in cell homeostasis, MCS dysfunction has been linked to several human diseases (*Area-Gomez et al., 2012*; *Castro et al., 2018*; *Stoica et al., 2014*). At the molecular level, depending on the contacting organelles [endoplasmic reticulum (ER)–Golgi, ER–mitochondria, ER–plasma membrane (PM), etc.], each MCS is enriched in specific proteinaceous factors that contribute to the specialized biological function of a given MCS (*Prinz et al., 2020*). By bridging the membrane of apposed organelles, either via protein–protein or protein–lipid interactions, MCS components also form tethering complexes that increase the affinity of one organelle to another and thereby keep their membranes in close proximity (*Eisenberg-Bord et al., 2016*; *Prinz et al., 2020*; *Scorrano et al., 2019*). Although the overall molecular composition of each MCS is different, one integral ER protein, the vesicle-associated membrane protein (VAMP)-associated protein (VAP) (*Murphy and Levine, 2016*), engages in tethering complexes at several MCS. This is accomplished by interaction of the cytosolic major sperm protein (MSP) domain of VAP with proteins containing two phenylalanine in an acidic tract (FFAT) motifs (*Loewen et al., 2003*; *Murphy and Levine, 2016*). FFAT motif-containing proteins include soluble proteins, such as lipid transfer proteins that contain an additional domain for targeting to the opposing membrane, and transmembrane proteins anchored to the contacting organelle (*James and Kehlenbach, 2021*). The molecular determinants driving the VAP–FFAT interaction have been investigated at the cellular and structural level. A consensus of the FFAT motif core was first defined as seven amino acids, $E^1F^2F^3D^4A^5x^6E^7$; however, the core motif of many identified VAP interacting proteins deviates from this canonical sequence (*James and Kehlenbach, 2021*; *Loewen et al., 2003*). In addition to the core, acidic residues surrounding the core motif are proposed to facilitate the VAP–FFAT interaction through electrostatic interactions (*Furuita et al., 2010*).

In addition to their critical role in interorganelle communication, MCS are exploited by intracellular pathogens for replication (*Derré, 2017*; *Ishikawa-Sasaki et al., 2018*; *Justis et al., 2017*; *McCune et al., 2017*). One example is the obligate intracellular bacterium *Chlamydia trachomatis*, the causative agent of the most commonly reported bacterial sexually transmitted infection. Upon invasion of the genital epithelium, *C. trachomatis* replicates within a membrane-bound vacuole called the inclusion (*Gitsels et al., 2019*). Maturation of the inclusion relies on *Chlamydia* effector proteins that are translocated across the inclusion membrane via a bacterially encoded Type III secretion system (*Lara-Tejero and Galán, 2019*). A subset of *Chlamydia* Type III effector proteins, known as the inclusion membrane proteins (Inc), are inserted into the inclusion membrane and are therefore strategically positioned to mediate inclusion interactions with host cell organelles (*Bugalhão and Mota, 2019*; *Dehoux et al., 2011*; *Lutter et al., 2012*; *Moore and Ouellette, 2014*). These interactions include points of contact between the ER and the inclusion membrane, without membrane fusion (*Derré et al., 2011*; *Dumoux et al., 2012*), which are referred to as ER-Inclusion MCS based on their similarities to MCS between cellular organelles (*Agaisse and Derré, 2015*; *Derré et al., 2011*).

Characterization of the protein composition of ER-Inclusion MCS led to the identification the Inc protein IncV, which constitutes a structural component that tethers the ER membrane to the inclusion membrane through interaction with VAP (*Stanhope et al., 2017*). The IncV–VAP interaction relies on the presence of two FFAT motifs in the C-terminal cytosolic tail of IncV. The core sequence of one of the motifs ($_{286}E^1Y^2M^3D^4A^5L^6E^7_{292}$) is similar to the canonical sequence, whereas a second motif ($_{262}S^1F^2H^3T^4P^5P^6N^7_{268}$) deviates significantly and was originally defined as a noncanonical FFAT (*Stanhope et al., 2017*). Similar to eukaryotic FFAT, the residue in position 2 in each motif ($Y_{287}$ and $F_{263}$, respectively) is essential for the IncV–VAP interaction during infection. However, it remains unclear whether additional determinants promote the assembly of this bacterial tether.

Here, we show that multiple layers of host cell kinase-mediated phosphorylation govern the assembly of the IncV–VAP tethering complex and ER-Inclusion MCS formation. IncV phosphorylation supports the IncV–VAP interaction through FFAT motifs displaying core domains immediately downstream of phosphorylation-mediated acidic tracts. Since the substitution for genetically encoded acidic tracts interfered with IncV translocation, we propose that *Chlamydia* evolved a post-translocation

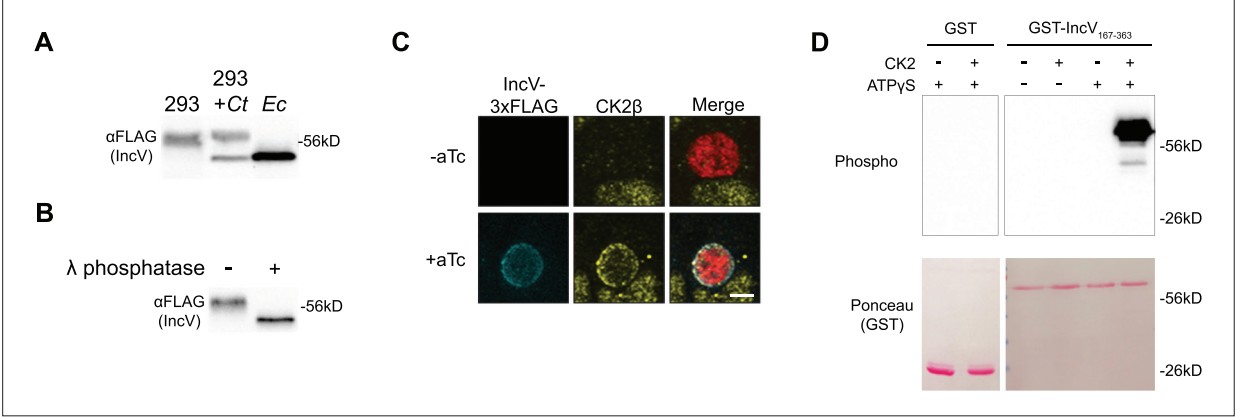

**Figure 1.** CK2 localizes to the inclusion and phosphorylates IncV. (**A**) Western blot of IncV-3xFLAG from lysates of HEK293 cells expressing IncV-3xFLAG (293), HEK293 cells infected with *C. trachomatis* expressing IncV-3xFLAG (293+*Ct*), or *E. coli* expressing IncV-3xFLAG (*Ec*). (**B**) Western blot of IncV-3xFLAG purified from lysates of HEK293 cells infected with *C. trachomatis* expressing IncV-3xFLAG and treated with lambda ($\lambda$) phosphatase (+) or phosphatase buffer alone (−). (**C**) Three-dimensional reconstruction of confocal images of HeLa cells infected with *C. trachomatis* expressing mCherry constitutively (red) and IncV-3xFLAG (blue) under the control of an anhydrotetracycline (aTc)-inducible promoter in the absence (−aTc) or presence (+aTc) of aTc and stained to detect endogenous CK2β (yellow). The merge is shown on the right. Scale bar is 5 μm. (**D**) In vitro kinase assay using GST or GST-IncV$_{167-363}$ purified from *E. coli* as a substrate in the presence (+) or absence (−) of recombinant CK2 and in the presence (+) or absence (−) of ATPγS. The top panel shows phosphorylated proteins detected with anti-thiophosphate antibodies and the bottom panel is the same membrane stained with Ponceau S to detect total proteins.

The online version of this article includes the following source data and figure supplement(s) for figure 1:

**Source data 1.** Uncropped, labeled blots for *Figure 1A*.

**Source data 2.** Uncropped, labeled blots for *Figure 1B*.

**Source data 3.** Uncropped, labelled blots for *Figure 1D*.

**Source data 4.** Raw data for FLAG blot in *Figure 1A*.

**Source data 5.** Raw data for FLAG blot in *Figure 1B*.

**Source data 6.** Raw data for thiophosphate blot 1 in *Figure 1D*.

**Source data 7.** Raw data for thiophosphate blot 2 in *Figure 1D*.

**Source data 8.** Raw data for Ponceau S blot 1 in *Figure 1D*.

**Source data 9.** Raw data for Ponceau S blot 2 in *Figure 1D*.

**Figure supplement 1.** IncV recruits CK2 to the inclusion membrane.

phosphorylation strategy in order to accommodate proper secretion via the Type III secretion system, while achieving full mimicry of eukaryotic FFAT motifs.

## Results

### IncV is modified by phosphorylation

When subjected to anti-FLAG western blot analysis, we noticed that lysates of HEK293 eukaryotic cells infected with wild-type *C. trachomatis* overexpressing IncV-3xFLAG displayed a doublet consisting of a 50 and 60 kDa band (*Figure 1A*, middle lane, 293+*Ct*). By contrast, IncV-3xFLAG ectopically expressed in HEK293 cells had an apparent molecular weight that was shifted toward the 60 kDa band of the doublet (*Figure 1A*, left lane, 293), while IncV-3xFLAG expressed in *E. coli* had an apparent molecular weight equivalent to the 50 kDa band of the doublet (*Figure 1A*, right lane, *Ec*). This result led us to hypothesize that IncV is post-translationally modified by a host factor.

To determine if phosphorylation could account for the increase in the apparent molecular weight of IncV, we performed a phosphatase assay. IncV-3xFLAG was immunoprecipitated, using anti-FLAG-conjugated Sepharose beads, from lysates of HEK293 cells infected with *C. trachomatis* expressing IncV-3xFLAG. Following the release of IncV-3xFLAG from the beads by FLAG peptide competition, the eluate was treated with lambda ($\lambda$) phosphatase or phosphatase buffer alone, and subsequently subjected to anti-FLAG western blot analysis (*Figure 1B*). In the absence of $\lambda$ phosphatase, the

apparent molecular weight of IncV-3xFLAG was approximately 60 kDa (*Figure 1B*, left lane). Upon phosphatase treatment, we observed a decrease in the apparent molecular weight of IncV-3xFLAG to approximately 50 kDa, similar to what was observed when IncV-3xFLAG was expressed in *E. coli* (*Figure 1B*, right lane). Altogether, these results demonstrate that overexpressed IncV is phosphorylated by a host cell kinase.

## The host kinase CK2 phosphorylates IncV

We next focused on identifying the host cell kinase(s) responsible for phosphorylating IncV. All three subunits of Protein Kinase CK2 were identified as potential interacting partners of IncV in an Inc-human interactome (*Mirrashidi et al., 2015*). To determine if CK2 associated with IncV at ER-Inclusion MCS, HeLa cells transfected with YFP-CK2α or YFP-CK2β constructs and infected with *C. trachomatis* wild-type expressing mCherry under a constitutive promoter and IncV-3xFLAG under the anhydro-tetracycline (aTc)-inducible promoter were analyzed by confocal immunofluorescence microscopy (*Figure 1—figure supplement 1*). In the absence of IncV-3xFLAG expression, YFP-CK2α and YFP-CK2β were undetectable at the inclusion (*Figure 1—figure supplement 1A, B*, −aTc). However, upon overexpression of IncV-3xFLAG, YFP-CK2α, and YFP-CK2β was recruited to the inclusion membrane and colocalized with IncV (*Figure 1—figure supplement 1A, B*, +aTc). To confirm that this phenotype was not the result of overexpression of the CK2 subunits, we used antibodies that recognized the endogenous CK2β subunit and showed that endogenous CK2β colocalized with IncV at the inclusion, when IncV-3xFLAG expression was induced (*Figure 1C*). Altogether, these results demonstrate that CK2 is a novel component of ER-Inclusion MCS that is recruited to the inclusion in an IncV-dependent manner.

Having established that IncV is phosphorylated and that CK2 localizes to ER-Inclusion MCS in an IncV-dependent manner, we next tested if CK2 phosphorylates IncV. We performed an in vitro kinase assay using recombinant CK2 and the cytosolic domain of IncV (amino acids 167–363 of IncV) fused to GST (GST-IncV$_{167-363}$) or GST alone, purified from *E. coli*. To detect phosphorylation, we used ATPγS, which can be utilized by kinases to thiophosphorylate a substrate, followed by an alkylation reaction of the thiol group to generate an epitope that is detected using an antibody that recognizes thio-phosphate esters (*Allen et al., 2007*). When GST alone was provided as a substrate, there was no detectable phosphorylation, regardless of the presence of CK2 and ATPγS (*Figure 1D*, lanes 1 and 2). A similar result was observed with GST-IncV$_{167-363}$ in the absence of CK2 and/or ATPγS (*Figure 1D*, lanes 3–5). However, in the presence of both ATPγS and CK2, GST-IncV$_{167-363}$ was phosphorylated (*Figure 1D*, lane 6). Altogether, these results demonstrate that CK2 directly phosphorylates IncV in vitro.

## Phosphorylation of IncV is necessary and sufficient to promote the IncV–VAP interaction in vitro

We have previously reported an IncV–VAP interaction in vitro upon incubation of IncV$_{167-363}$ with the cytosolic MSP domain of VAP (GST-VAP$_{MSP}$) purified from *E. coli* (*Stanhope et al., 2017*). However, this interaction was only detected when IncV$_{167-363}$ was produced in eukaryotic cells, which, based on the above results, led us to hypothesize that IncV phosphorylation is required for the IncV–VAP interaction. We assessed the role of phosphorylation in the IncV–VAP interaction by performing lambda ($\lambda$) phosphatase dephosphorylation of IncV coupled with a GST-VAP$_{MSP}$ pull-down assay (*Figure 2A*). IncV-3xFLAG was immunoprecipitated from lysates of HEK293 cells using anti-FLAG-conjugated Sepharose beads, released from the beads using FLAG peptide competition, and treated with $\lambda$ phosphatase or buffer alone. Treated and untreated IncV-3xFLAG samples were then incubated with GST-VAP$_{MSP}$ or GST alone bound to glutathione Sepharose beads. The protein-bound beads were subjected to western blot analysis using an anti-FLAG antibody (*Figure 2B*). Untreated IncV-3xFLAG was pulled down by GST-VAP$_{MSP}$ but not by GST alone, demonstrating a specific interaction between IncV and VAP (*Figure 2B*, lanes 1–3). However, when the eluate containing IncV-3xFLAG was treated with $\lambda$ phosphatase prior to incubation with GST-VAP$_{MSP}$, the two proteins failed to interact (*Figure 2B*, lane 4), indicating that phosphorylation of IncV is necessary for the IncV–VAP interaction in vitro.

We next determined if IncV phosphorylation by CK2 was sufficient to promote the IncV–VAP interaction in an in vitro binding assay (*Figure 2C*). MBP-tagged VAP$_{MSP}$ (MBP-VAP$_{MSP}$) and GST-IncV$_{167-363}$ were expressed separately in *E. coli* and purified using amylose resin and glutathione Sepharose

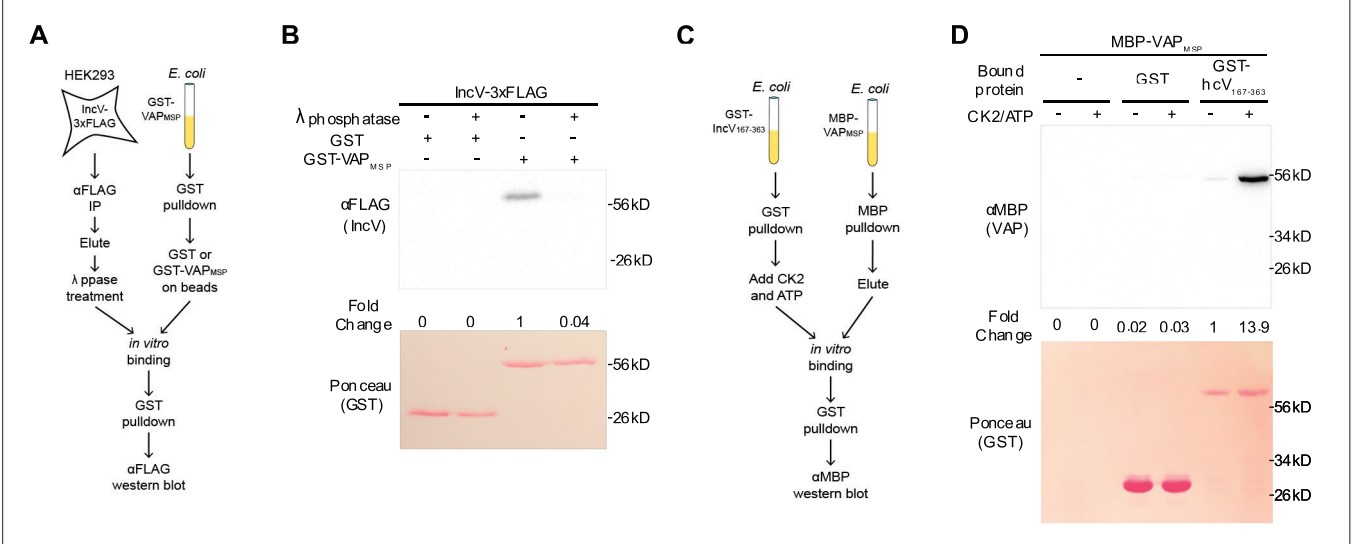

**Figure 2.** Phosphorylation of IncV is necessary and sufficient to promote the IncV–VAP interaction in vitro. (**A**) Schematic depicting the experimental setup for results in B. (**B**) In vitro binding assay using IncV-3xFLAG purified from HEK293 lysates and treated with lambda ($\lambda$) phosphatase (+) or phosphatase buffer alone (−) combined with GST or GST-VAP$_{MSP}$ purified from *E. coli* and immobilized on glutathione beads. The top panel shows proteins detected with anti-FLAG antibodies and the bottom panel is the same membrane stained with Ponceau S to detect total protein. (**C**) Schematic depicting the experimental setup for results in D. (**D**) In vitro binding assay using GST or GST-IncV$_{167–363}$ purified from *E. coli*, and immobilized on glutathione beads, as a substrate for CK2 in the presence (+) or absence (−) of CK2 and ATP, combined with MBP-VAP$_{MSP}$ purified from *E. coli*. The top panel was probed with anti-MBP and the bottom panel was the same membrane stained with Ponceau S to detect the GST construct.

The online version of this article includes the following source data for figure 2:

**Source data 1.** Quantification of blot densities for *Figure 2*.

**Source data 2.** Uncropped, labeled blots for *Figure 2B*.

**Source data 3.** Uncropped, labelled blots for *Figure 2D*.

**Source data 4.** Raw data for FLAG blot in *Figure 2B*.

**Source data 5.** Raw data for Ponceau S blot in *Figure 2B*.

**Source data 6.** Raw data for MBP blot in *Figure 2D*.

**Source data 7.** Raw data for Ponceau S blot in *Figure 2D*.

beads, respectively. GST-IncV$_{167–363}$ was left attached to glutathione Sepharose beads and was phosphorylated by incubation with recombinant CK2 and ATP before being combined with purified MBP-VAP$_{MSP}$. GST-IncV$_{167–363}$ was pulled down and the samples were subjected to western blot using anti-MBP antibodies (*Figure 2D*). Neither the beads alone, nor GST alone pulled down MBP-VAP$_{MSP}$, regardless of whether CK2 and ATP were present or not (*Figure 2D*, lanes 1–4). In the absence of CK2 and ATP, we observed minimal binding of MBP-VAP$_{MSP}$ to GST-IncV$_{167–363}$ (*Figure 2D*, lane 5). However, when GST-IncV$_{167–363}$ was treated with CK2 and ATP prior to GST-pull-down, MBP-VAP$_{MSP}$ and GST-IncV$_{167-363}$ co-immuno-precipitated, indicating that phosphorylation of IncV by CK2 is sufficient to promote the IncV–VAP interaction in vitro (*Figure 2D*, lane 6). Altogether, these results demonstrate that IncV phosphorylation is necessary and sufficient for the IncV–VAP interaction in vitro.

## CK2 kinase activity is required for IncV phosphorylation and IncV–VAP interaction at the inclusion

We next determined the contribution of CK2 to IncV phosphorylation and the subsequent assembly of the IncV–VAP tether at the inclusion. We first used a genetic approach to deplete CK2β. Because CK2 is essential (*Buchou et al., 2003*; *Zhang et al., 2004*), we favored a gene silencing approach over a CRISPR-based knockout approach. HeLa cells treated with individual siRNA duplexes targeting *CSNK2B* (A, B, C, or D), or a pool of all four siRNA duplexes (pool), were infected with a previously characterized *incV* mutant strain of *C. trachomatis* (*Stanhope et al., 2017*; *Weber et al., 2017*), expressing IncV$_{WT}$-3xFLAG from an aTc-inducible promoter. The cells were lysed and subjected to

western blot analysis. The efficacy of *CSNK2B* knockdown was confirmed by western blot, demonstrating that, in siRNA treated cells, CK2β protein levels ranged from 9.3% to 53.3% compared to control cells (*Figure 3A*, middle blot). As shown in *Figure 1A*, in control cells, IncV$_{WT}$-3xFLAG appeared as a doublet (*Figure 3A*, top blot, left lane, ooo and o). In contrast, depletion of CK2β led to the appearance of additional bands of intermediate apparent molecular weight (*Figure 3A*, top and middle blots, pool, A, B, C, D lanes, oo). A line scan analysis of the control sample revealed two peaks corresponding to the top band, corresponding to hyperphosphorylated IncV (*Figure 3B*, black line, left peak, ooo) and to the bottom band, corresponding to unphosphorylated IncV (*Figure 3B*, black line, right peak, o). A similar analysis of the banding pattern of IncV upon CK2β depletion, with the pooled or individual siRNA duplexes, revealed the appearance of intermediate peaks between the top and bottom bands, suggesting the formation of hypophosphorylated species of IncV (*Figure 3B*, middle peaks, oo). These results provided a first indication that CK2 mediates IncV phosphorylation during infection. However, none of the siRNA duplex treatments led to a complete dephosphorylation of IncV, which could be due to the incomplete knockdown of CK2β (*Figure 3A*, middle blot).

To complement the genetic approach described above, we conducted a pharmacological approach using the CK2-specific inhibitor CX-4945 (*Rusin et al., 2017*). HeLa cells infected with a *C. trachomatis incV* mutant expressing IncV$_{WT}$-3xFLAG under the control of the aTc-inducible promoter were treated with increasing concentrations of CX-4945 (0, 0.625, and 10 μM) at 18 hours post infection (h pi), prior to the induction of IncV$_{WT}$-3xFLAG expression at 20 h pi. This experimental setup allowed for CK2 inhibition, prior to IncV$_{WT}$-3xFLAG synthesis, translocation, insertion into the inclusion membrane and exposure to the host cell cytosol. The cells were lysed 24 h pi and subjected to western blot analysis to determine the effect of CK2 inhibition on the apparent molecular weight of IncV. The apparent molecular weight of IncV decreased in a dose-dependent manner (*Figure 3C*, top blot), leading to an apparent molecular weight corresponding to unphosphorylated IncV at the 10 μM concentration. These results demonstrate that CK2 activity is essential for IncV phosphorylation during infection.

We next determined whether inhibition of CK2 kinase activity affected the IncV-dependent VAP recruitment to the inclusion and, therefore, the assembly of the IncV–VAP tether. We used the same experimental setup as above, except that the cells expressed CFP-VAP. At 24 h pi, the cells were fixed, immunostained with anti-FLAG antibody, and processed for confocal microscopy. Qualitative and quantitative assessment of the micrographs indicated that CX-4945 did not interfere with IncV translocation and insertion into the inclusion membrane (*Figure 3D* and *Figure 3—figure supplement 1*). The quantification method is illustrated in *Figure 3—figure supplement 2*. As previously observed (*Stanhope et al., 2017*), IncV$_{WT}$-3xFLAG overexpression correlated with a strong CFP-VAP association with the inclusion (*Figure 3D*, top panels). In comparison, pretreatment of the cells with 10 μM of CX-4945 abolished VAP recruitment to the inclusion (*Figure 3D*, bottom panels). Quantification of the CFP-VAP signal associated with IncV at the inclusion membrane confirmed the qualitative analysis and also revealed an intermediate phenotype for cells treated with 0.625 μM of CX-4945 (*Figure 3D, E*). To rule out any off-target effect of CX-4945, we tested an independent CK2-specific inhibitor, GO289 (*Borgo et al., 2021*). As observed with CX-4945, GO289-mediated CK2 inhibition led to the dose-dependent dephosphorylation of IncV (*Figure 3—figure supplement 3A*) and a significant reduction in the percentage of inclusions associated with VAP (*Figure 3—figure supplement 3B, C*). Altogether, these results demonstrate that phosphorylation of IncV by CK2 is required for the IncV-dependent VAP recruitment to the inclusion.

## CK2 kinase activity is dispensable for CK2 localization to the inclusion and reversing CK2 inhibition restores VAP recruitment to the inclusion

To further investigate the sequence of events leading to the CK2- and IncV phosphorylation-dependent VAP recruitment to the inclusion, we first determined if CK2 kinase activity was required for CK2 recruitment to the inclusion. HeLa cells expressing YFP-CK2β were infected with the *C. trachomatis incV* mutant expressing IncV$_{WT}$-3xFLAG. 10 μM CX-4945 was added, or not, at 18 h pi, and IncV expression was induced at 20 h pi by addition of aTc, as described in *Figure 3D, E*. At 24 h pi, the cells were fixed, immunostained with anti-FLAG antibody, and processed for confocal microscopy. Qualitative and quantitative assessment of the micrographs indicated that CX-4945 did not interfere with CK2 association with the inclusion (*Figure 4A, B*). Thus, CK2 kinase activity is dispensable for the IncV-dependent CK2 localization to the inclusion.

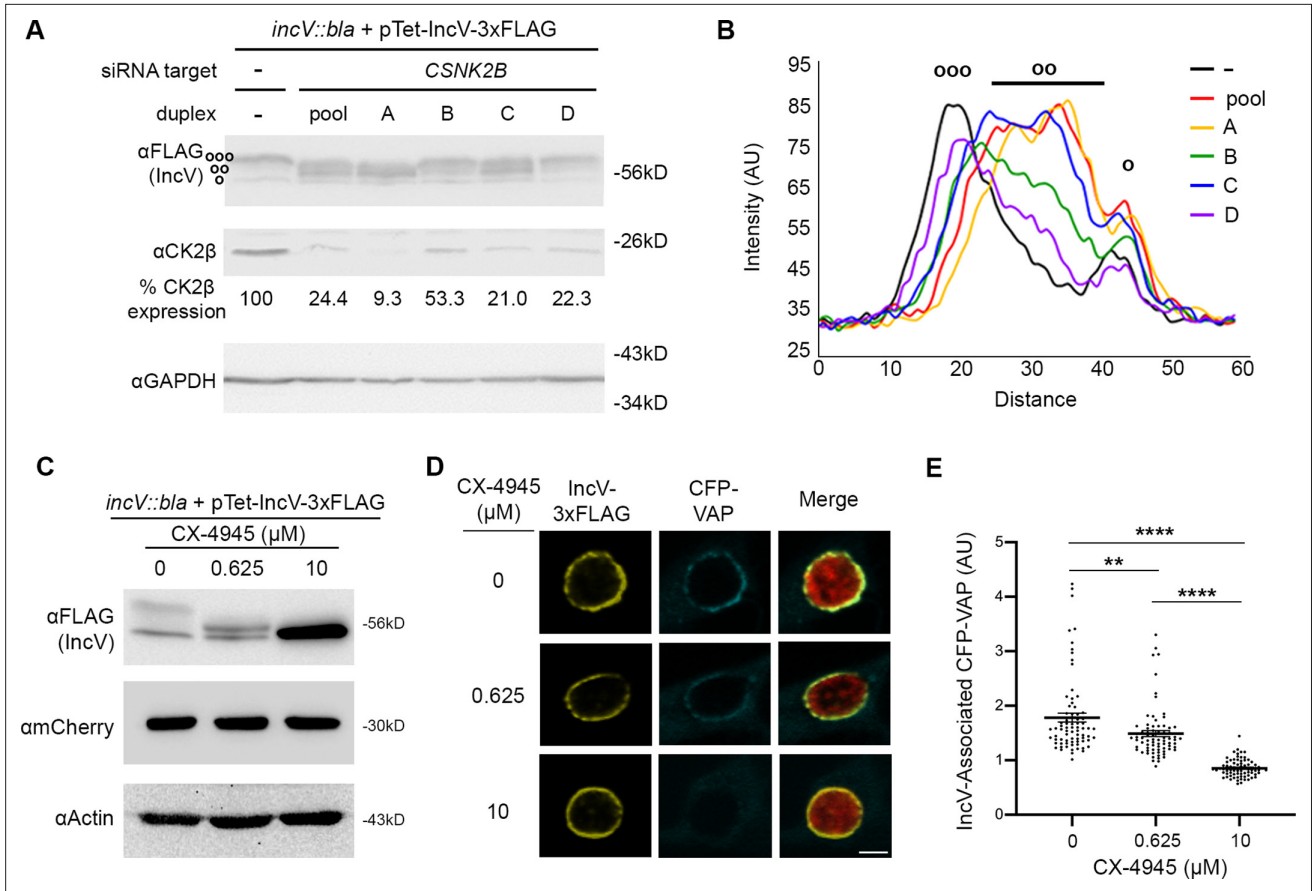

**Figure 3.** CK2 plays a role in the IncV–VAP interaction during infection. (**A**) Western blot of lysates of HeLa cells treated with siRNA buffer alone (−) or with siRNA duplexes targeting *CSNK2B* (pool of four duplexes or individual duplexes A, B, C, or D) and infected with a *C. trachomatis incV* mutant expressing IncV-3xFLAG. The top panel was probed with anti-FLAG. The middle panel was probed with anti-CK2β. The bottom panel was probed with anti-GAPDH. Relative expression levels of CK2β normalized to GAPDH loading controls are shown as a percentage of no siRNA control expression. (ooo) hyperphosphorylated IncV, (oo) intermediate hypophosphorylated IncV, and (o) unphosphorylated IncV. (**B**) Line scan analysis of FLAG signal detected in A. The peak on the left (ooo) corresponds to the hyperphosphorylated species of IncV, and the peak on the right (o) corresponds to the unphosphorylated species of IncV. Intermediate hypophosphorylated species are indicated by any peak between the left and right peaks (oo). Each line represents a different condition: Control, black; siRNA pool of duplexes A–D, red; siRNA duplex A, yellow; siRNA duplex B, green; siRNA duplex C, blue; siRNA duplex D, purple. (**C–E**) HeLa cells, expressing CFP-VAP (D and E only), were infected with *C. trachomatis incV* mutant expressing IncV-3xFLAG under the control of the anhydrotetracycline (aTc)-inducible promoter and treated with increasing concentrations of the CK2 inhibitor CX-4945 (0, 0.625, 10 µM) for 2 hr at 18 h pi and prior to the induction of IncV-3xFLAG expression at 20 h pi. The samples were processed 24 h pi for western blot (C) or confocal microscopy (D, E). (**C**) Cell lysates were probed with anti-FLAG (top blot), anti-mCherry (middle blot), or anti-actin (bottom blot) antibodies. (**D**) Single plane confocal micrographs of HeLa cells expressing CFP-VAP (blue), infected with *incV* mutant expressing IncV-3xFLAG (yellow) and mCherry (red). The merge is shown on the right. Scale bar is 5 µm. (**E**) Quantification of the mean intensity of the CFP-VAP signal within an object generated from the IncV-3xFLAG signal and normalized to the mean intensity of CFP-VAP in the ER. Each dot represents one inclusion. Data show the mean and SEM of a combination of three independent experiments. One-way ANOVA and Tukey's post hoc test were performed. $**p < 0.01$, $****p < 0.0001$.

The online version of this article includes the following source data and figure supplement(s) for figure 3:

**Source data 1.** Quantification of blot densities, line scan analysis, and IncV-associated CFP-VAP for *Figure 3*.

**Source data 2.** Uncropped, labeled blots for *Figure 3A*.

**Source data 3.** Uncropped, labelled blots for *Figure 3C*.

**Source data 4.** Raw data for FLAG blot in *Figure 3A*.

**Source data 5.** Raw data for CK2 blot in *Figure 3A*.

**Source data 6.** Raw data for GAPDH blot in *Figure 3A*.

**Source data 7.** Raw data for FLAG blot in *Figure 3C*.

**Source data 8.** Raw data for mCherry blot in *Figure 3C*.

*Figure 3 continued on next page*

*Figure 3 continued*

**Source data 9.** Raw data for Actin blot in *Figure 3C*.

**Figure supplement 1.** IncV inclusion localization is not affected upon CX-4945 treatment.

**Figure supplement 1—source data 1.** Quantification of inclusion-associated IncV for *Figure 3—figure supplement 1*.

**Figure supplement 2.** Method to quantify the inclusion association of a given marker.

**Figure supplement 3.** The CK2 kinase inhibitor GO289 inhibits IncV phosphorylation and VAP recruitment to the inclusion.

**Figure supplement 3—source data 1.** Uncropped, labelled blots for *Figure 3—figure supplement 3A*.

**Figure supplement 3—source data 2.** Raw data for FLAG blot in *Figure 3—figure supplement 3A*.

**Figure supplement 3—source data 3.** Raw data for mCherry blot in *Figure 3—figure supplement 3A*.

**Figure supplement 3—source data 4.** Raw data for Actin blot in *Figure 3—figure supplement 3A*.

**Figure supplement 3—source data 5.** Quantification of percentage of CFP-VAP-positive inclusions in *Figure 3—figure supplement 3C*.

Altogether, our results suggest that CK2 is recruited first and that VAP recruitment only occurs once CK2 phosphorylates IncV. To test this model, we determined if reversing CK2 inhibition, after CK2 had been recruited to the inclusion, would restore VAP association with the inclusion (*Figure 4C*). HeLa cells expressing YFP-CK2β and CFP-VAP, and infected with the *C. trachomatis incV* mutant expressing IncV$_{WT}$-3xFLAG, were treated with CX-4945 at 18 h pi. *incV* expression was induced at 20 h pi by addition of aTc, in the presence of CX-4945. At 24 h pi, the cells were washed and incubated with media containing either aTc and CX-4945 (CX-4945 replaced) or aTc only (CX-4945 washout) for an additional hour (25 h pi). Samples where CX-4945 was omitted and collected at 25 h pi (No CX-4945), and samples processed in the presence of CX-4945 and collected at 24 h pi prior to the wash (CX-4945) were used as controls. None of the treatment prevented IncV localization to the inclusion (*Figure 4—figure supplement 1*). Qualitative and quantitative assessment of the micrographs confirmed that, as observed before, CX-4945 had no effect on CK2 association with the inclusion but strongly prevented VAP recruitment (*Figure 4D, E*, compare CX-4945 to No CX-4945). A similar result was observed when the media was replenished with CX-4945 after the wash (*Figure 4D, E*, compare CX-4945 replaced to CX-4945). In comparison, CX-4945 washout post CK2 association with the inclusion, led to a robust VAP recruitment to the inclusion, similar to the one observed in the absence of CX-4945 (*Figure 4D, E*, compare CX-4945 washout to No CX-4945). Altogether, these results indicate that the IncV-dependent CK2 association with the inclusion does not require CK2 kinase activity and precedes VAP recruitment, which requires an active CK2 kinase.

## Three serine residues in a C-terminal domain of IncV control CK2 and VAP recruitment to the inclusion, IncV hyperphosphorylation and ER-Inclusion MCS formation

To gain further mechanistic insight about the CK2–IncV–VAP interplay, and its role in ER-Inclusion MCS formation, we next determined which domain of IncV was important for the recruitment of CK2 to the inclusion by generating a series of C-terminal truncated IncV constructs (*Figure 5A*). These constructs, as well as the full-length IncV (FL, 1–363), were cloned under the aTc-inducible promoter and expressed from the *C. trachomatis incV* mutant strain. All IncV constructs similarly localized to the inclusion membrane (*Figure 5—figure supplement 1A*). HeLa cells expressing YFP-CK2β were infected with each of the complemented strains, and the ability of the truncated versions of IncV to recruit YFP-CK2β to the inclusion was assessed by confocal microscopy. Qualitative and quantitative analysis revealed that, compared to full-length IncV$_{FL}$-3xFLAG, IncV$_{1–341}$-3xFLAG was no longer capable of recruiting YFP-CK2β to the inclusion, whereas IncV$_{1–356}$-3xFLAG was moderately affected (*Figure 5—figure supplement 2A, B*). Additionally, strains expressing IncV$_{1–341}$-3xFLAG also exhibited a significant reduction in IncV-associated VAP compared to IncV$_{FL}$- or IncV$_{1–356}$-3xFLAG (*Figure 5—figure supplement 2C, D*). Altogether, these results demonstrate that a C-terminal region of IncV, between amino acids 342 and 356, is required for the IncV-dependent CK2 recruitment to the inclusion and subsequent VAP association with the inclusion.

The primary amino acid structure of the IncV domain necessary for CK2 recruitment ($_{342}$SSESS-DEESSSDSS$_{356}$) contains seven CK2 recognition motifs (S/T-x-x-D/E/pS/pY) (*Litchfield, 2003*; *Figure 5A*). Three of them do not require priming by phosphorylation of the fourth serine or tyrosine

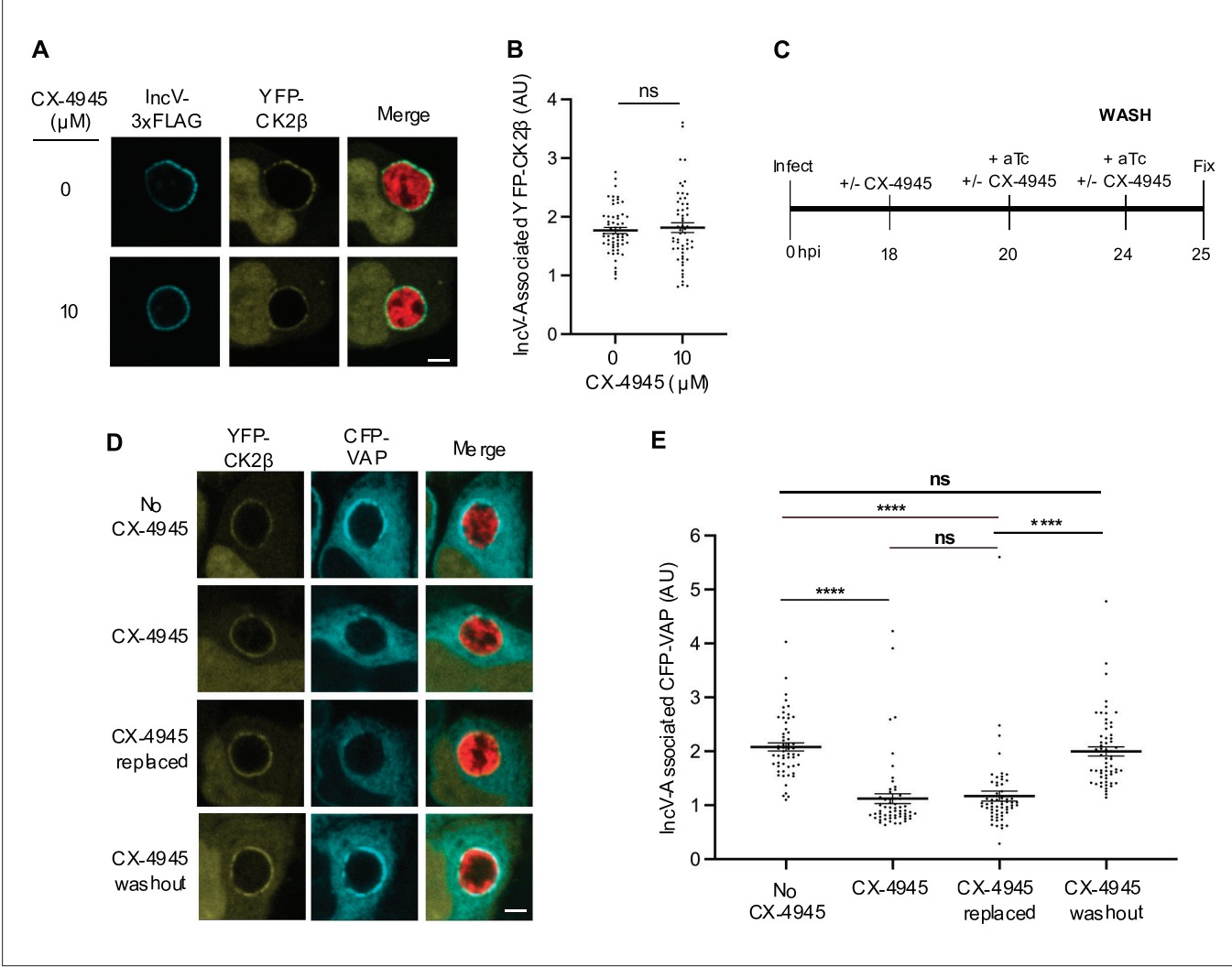

**Figure 4.** CK2 kinase activity is dispensable for CK2 localization to the inclusion and reversing CK2 inhibition allows for VAP recruitment. (**A**) Single plane confocal micrographs of HeLa cells expressing YFP-CK2β (yellow), infected with an *incV* mutant expressing IncV-3xFLAG (blue) and mCherry (red), and treated with 10 μM CX-4945 (bottom panels), or not (top panels), for 2 hr at 18 h pi and prior to the induction of IncV-3xFLAG expression at 20 h pi. The merge is shown on the right. Scale bar is 5 μm. (**B**) Quantification of the mean intensity of the YFP-CK2β signal within an object generated from the IncV-3xFLAG signal and normalized to the mean intensity of YFP-CK2β in the cytosol. Each dot represents one inclusion. Data show the mean and SEM of a combination of three independent experiments. ns: nonsignificant (Student's *t*-test). (**C**) Schematic depicting the experimental setup for the results in D and E. (**D**) Single plane confocal micrographs of HeLa cells expressing CFP-VAP (blue) and YFP-CK2β (yellow), infected with an *incV* mutant expressing IncV-3xFLAG under the control of the anhydrotetracycline (aTc)-inducible promoter and mCherry (red). Infections were performed in the absence of CX-4945 (No CX-4945) or presence of 10 μM CX-4945 for the duration of the experiment (CX-4945). Alternatively, CX-4945 was present until 24 h pi, when the cells were washed and incubated with media containing either aTc and CX-4945 (CX-4945 replaced) or aTc only (CX-4945 washout) for an additional hour. The merge is shown on the right. Scale bar is 5 μm. (**E**) Quantification of the mean intensity of the CFP-VAP signal within an object generated from the YFP-CK2β signal and normalized to the mean intensity of CFP-VAP in the ER. Each dot represents one inclusion. Data show the mean and SEM of a combination of three independent experiments. One-way ANOVA and Tukey's post hoc test were performed. ****p < 0.0001.

The online version of this article includes the following source data and figure supplement(s) for figure 4:

**Source data 1.** Quantification of IncV-associated YFP-CK2β and IncV-associated CFP-VAP for *Figure 4*.

**Figure supplement 1.** IncV inclusion localization is not affected upon CX-4945 replacement/washout treatment.

**Figure supplement 1—source data 1.** Quantification of inclusion-associated IncV for *Figure 4—figure supplement 1*.

residue and could result in the direct CK2-dependent phosphorylation of IncV on serine residues $S_{345}$, $S_{346}$, and $S_{350}$, hereby facilitating the assembly of the IncV–VAP tether. To test this hypothesis, all three serine residues were substituted for unphosphorylatable alanine residues (IncV$_{S345A-S346A-S350A}$ referred to as IncV$_{S3A}$). HeLa cells expressing YFP-CK2β or YFP-VAP were infected with *C. trachomatis incV*

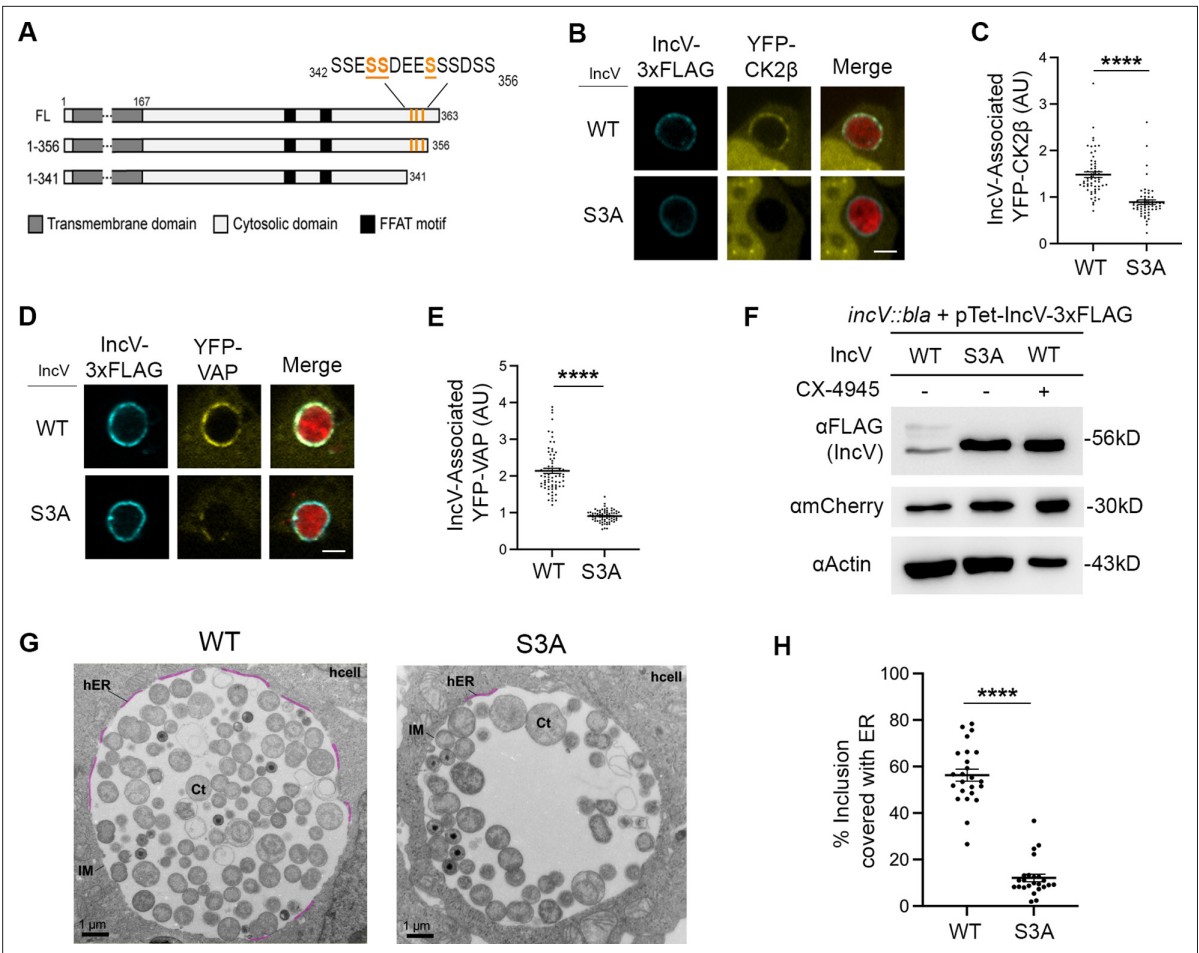

**Figure 5.** Three serine residues in a C-terminal domain of IncV mediate CK2 and VAP recruitment to the inclusion, IncV phosphorylation, and ER-Inclusion membrane contact site (MCS) formation. (**A**) Schematic depicting truncated IncV constructs. The numbers indicate the amino acid position within the IncV protein sequence. CK2 phosphorylation sites that do not require priming are indicated in orange. Single plane confocal images of HeLa cells expressing YFP-CK2β (**B**) or YFP-VAP (**D**) (yellow), infected with a *C. trachomatis incV* mutant expressing mCherry (red) and IncV$_{WT}$– (WT) or IncV$_{S345A/S346A/S350A}$-3xFLAG (S3A) (blue). The merge is shown on the right. Scale bar is 5 µm. Quantification of the mean intensity of YFP-CK2β (**C**) and YFP-VAP (**E**) within the IncV object normalized to the mean intensity of YFP-CK2β in the cytosol and YFP-VAP in the ER, respectively. Data show the mean and SEM of a combination of three independent experiments. ****p < 0.0001 (Student's *t*-test). (**F**) Western blot of lysates of HeLa cells infected with a *C. trachomatis incV* mutant expressing IncV$_{WT}$-3xFLAG (WT), IncV$_{S3A}$-3xFLAG (S3A), or expressing IncV$_{WT}$-3xFLAG and treated with 10 µM CX-4945 as described in *Figure 3C* (WT; CX-4945 +) and probed with anti-FLAG (top blot), anti-mCherry (middle blot), and anti-actin (bottom blot) antibodies. (**G**) Transmission electron micrographs of sections of HeLa cells infected with a *C. trachomatis incV* mutant expressing IncV$_{WT}$– (WT) or IncV$_{S3A}$-3xFLAG (S3A). Representative sections across a whole inclusion are shown. ER-Inclusion MCS are highlighted in pink. Corresponding images without the highlighted MCS are shown in *Figure 5—figure supplement 3C, D*. Ct: *Chlamydia trachomatis*; IM: inclusion membrane; hER: host endoplasmic reticulum; hcell: host cell cytosol. Scale bars are 1 µm. (**H**) Quantification of the percentage of the inclusion membrane covered with host ER. Each dot represents one section across a whole inclusion. Data show the mean and SEM for 24 representative electron micrographs per condition. ****p < 0.0001 (Student's *t*-test).

The online version of this article includes the following source data and figure supplement(s) for figure 5:

**Source data 1.** Quantification of IncV-associated YFP-CK2β and IncV-associated CFP-VAP for *Figure 5*.

**Source data 2.** Uncropped, labeled blots for *Figure 5F*.

**Source data 3.** Raw data for FLAG and mCherry blots in *Figure 5F*.

**Source data 4.** Raw data for Actin blot in *Figure 5F*.

**Source data 5.** Quantification of percentage of inclusion membrane associated with host ER for *Figure 5H*.

**Figure supplement 1.** Truncation and alanine substitution of S245, S346, and S350 does not affect IncV localization to the inclusion membrane.

**Figure supplement 1—source data 1.** Quantification of inclusion-associated IncV for *Figure 5—figure supplement 1*.

**Figure supplement 2.** A C-terminal domain of IncV mediates VAP recruitment to the inclusion.

*Figure 5 continued on next page*

*Figure 5 continued*

**Figure supplement 2—source data 1.** Quantification of IncV-associated YFP-CK2β and IncV-associated CFP-VAP for *Figure 5—figure supplement 2*.

**Figure supplement 3.** Ultrastructural analysis of ER-Inclusion membrane contact site (MCS).

**Figure supplement 4.** Phosphomimetic mutation of three serine residues in the C-terminal domain of IncV is not sufficient to promote the IncV–VAP interaction in vitro.

**Figure supplement 4—source data 1.** Quantification of blot densities for *Figure 5—figure supplement 4B*.

**Figure supplement 4—source data 2.** Uncropped, labeled blots for *Figure 5—figure supplement 4B*.

**Figure supplement 4—source data 3.** Raw data for MBP blot in *Figure 5—figure supplement 4B*.

**Figure supplement 4—source data 4.** Raw data for Ponceau S blot in *Figure 5—figure supplement 4B*.

mutant strains expressing IncV$_{WT}$- or IncV$_{S3A}$-3xFLAG. The cells were fixed at 24 h pi, immunostained with anti-FLAG antibody, and analyzed by confocal immunofluorescence microscopy. IncV$_{WT}$- and IncV$_{S3A}$-3xFLAG displayed similar inclusion localization (*Figure 5—figure supplement 1B*). However, qualitative and quantitative analysis revealed that in comparison to IncV$_{WT}$-3xFLAG, IncV$_{S3A}$-3xFLAG expression resulted in a significant decrease in both CK2β and VAP recruitment to the inclusion (*Figure 5B–E*). Altogether, these results indicate that serine residues S$_{345}$, S$_{346}$, and S$_{350}$ located in a C-terminal motif of IncV, are critical for CK2 and VAP recruitment to the inclusion.

To determine if IncV$_{S3A}$ failed to recruit VAP to the inclusion because of a lack of IncV phosphorylation, we assessed IncV$_{S3A}$ apparent molecular weight by western blot analysis of lysates from HeLa cells infected with a *C. trachomatis incV* mutant expressing IncV$_{WT}$- or IncV$_{S3A}$-3xFLAG. Compared to IncV$_{WT}$-3xFLAG, which as previously observed ran as a doublet corresponding to both phosphorylated and unphosphorylated species of IncV (*Figure 5F*, lane 1), the apparent molecular weight of IncV$_{S3A}$-3xFLAG (*Figure 5F*, lane 2), was identical to that of unphosphorylated IncV$_{WT}$-3xFLAG upon treatment with the CK2 inhibitor CX-4945 (*Figure 5F*, lane 3). These results indicated that IncV$_{S3A}$ is unphosphorylated.

To more directly demonstrate that, in absence of IncV hyperphosphorylation, the lack of VAP association with the inclusion membrane correlates with a defect in ER-Inclusion MCS formation, cells infected with a *C. trachomatis incV* mutant expressing IncV$_{WT}$- or IncV$_{S3A}$-3xFLAG were processed for electron microscopy. Qualitative assessment of the micrographs revealed that the integrity of the inclusion membrane and ER-Inclusion MCS were preserved in both conditions, based on the intermembrane distance of 10–20 nm between the ER tubules and the inclusion membrane, and on the presence of ribosomes on the cytosolic face of the ER tubules (*Figure 5—figure supplement 3A, B*). However, 61.9% (39/63) of the S3A inclusion sections displayed no ER in association with the inclusion membrane, compared to only 6.6% (4/61) of the WT inclusion sections. Additionally, quantification of the proportion of inclusion membrane associated with the ER indicated a significant reduction of ER-Inclusion MCS formation upon IncV$_{S3A}$ expression, compared to IncV$_{WT}$ (*Figure 5G, H* and *Figure 5—figure supplement 3C, D*). We note that some levels of ER-Inclusion MCS are expected with IncV$_{S3A}$ because of redundant tethering mechanism(s) (*Stanhope et al., 2017*). Altogether, these results indicate that the CK2-dependent phosphorylation of IncV controls assembly of the IncV–VAP tether and ER-Inclusion MCS formation.

Finally, we determined if phosphorylation of S$_{345}$, S$_{346}$, and S$_{350}$ was sufficient to mediate the in vitro IncV–VAP interaction observed upon CK2 phosphorylation of IncV (*Figure 1D*), by substituting S$_{345}$, S$_{346}$, and S$_{350}$ for phosphomimetic aspartic acid residues. The corresponding IncV construct, referred to as IncV$_{S3D}$, was purified from *E. coli* and tested for VAP binding in vitro. IncV$_{S3D}$ did not result in a significant increase in VAP binding compared to IncV$_{WT}$ (*Figure 5—figure supplement 4*). While we cannot exclude that the phosphomimetic mutations failed to mimic phosphorylation, these results indicated that, although critical for CK2 recruitment, IncV hyperphosphorylation status, and assembly of the IncV–VAP tether at the ER-inclusion MCS, phosphorylation of S$_{345}$, S$_{346}$, and S$_{350}$ alone is not sufficient to promote VAP binding in vitro, suggesting that additional IncV phosphorylation sites are required to promote optimal interaction between IncV and VAP.

## Phosphorylation of $T_{265}$ in the noncanonical FFAT of IncV contributes to the IncV–VAP interaction

We have previously shown that IncV displays one noncanonical ($_{262}S^1F^2H^3T^4P^5P^6N^7_{268}$) and one canonical FFAT motif ($_{286}E^1Y^2M^3D^4A^5L^6E^7_{292}$) (*Figure 6A*; *Stanhope et al., 2017*). In agreement with position 2 of a FFAT motif being a phenylalanine or a tyrosine residue critical for VAP–FFAT interactions (*Kawano et al., 2006*; *Loewen et al., 2003*), we had shown that alanine substitution of residue in position 2 of each motif, individually (IncV$_{F263A}$ or IncV$_{Y287A}$) and in combination (IncV$_{F263A/Y287A}$), led to a partial and full reduction of the IncV–VAP interaction, respectively, indicating that both FFAT motifs cooperate for VAP binding (*Stanhope et al., 2017*). Recently, Di Mattia et al. identified a new class of FFAT motifs referred to as phospho-FFAT motifs in which the acidic residue in position 4 is replaced by a phosphorylatable residue, such as serine or threonine (*Di Mattia et al., 2020*). The presence of a phosphorylatable threonine residue in position 4 of the noncanonical FFAT motif of IncV ($T_{265}$) (*Figure 6A*) suggests that, as proposed by Di Mattia et al., the noncanonical FFAT of IncV could be a phospho-FFAT motif. To test this hypothesis, $T_{265}$ was substituted for an alanine residue either individually (IncV$_{T265A}$), or in combination with alanine mutation of the tyrosine residue in position two of the canonical FFAT (IncV$_{T265A/Y287A}$). In parallel, a phosphomimetic mutation was generated by substituting both $T_{265}$ to an aspartic acid residue and the proline residue at position 266 ($P_{266}$) to an alanine residue, as described by *Di Mattia et al., 2020*, either individually (IncV$_{T265D/P266A}$) or in combination with a mutation in the canonical FFAT (IncV$_{T265D/P266A/Y287A}$). HeLa cells expressing YFP-VAP were infected with the *incV* mutant strain of *C. trachomatis* expressing IncV$_{T265A}$-, IncV$_{T265A/Y287A}$-, IncV$_{T265D/P266A}$-, or IncV$_{T265D/P266A/Y287A}$-3xFLAG. Cells infected with *incV* mutant strains expressing IncV$_{WT}$-, IncV$_{Y287A}$-, or IncV$_{F263A/Y287A}$-3xFLAG were included as controls. The cells were fixed at 24 h pi and immunostained with anti-FLAG antibody, followed by qualitative and quantitative analysis of confocal immunofluorescence microscopy images (*Figure 6B, C*). Although a few of the IncV constructs displayed statistically significant decreases in inclusion localization (*Figure 6—figure supplement 1A*), the reduction was minor, leaving a substantial amount of IncV on the inclusion membrane (84–89% of the WT on average) and most likely accounts for very little, if any, to the phenotypes described throughout our study. As previously observed (*Stanhope et al., 2017*), IncV$_{WT}$ exhibited a strong association of YFP-VAP with the inclusion membrane, while IncV$_{Y287A}$ and IncV$_{F263A/Y287A}$ exhibited a significant partial and full loss of inclusion-associated YFP-VAP, respectively (*Figure 6B, C*). Similarly, IncV$_{T265A}$ and IncV$_{T265A/Y287A}$ exhibited partial and complete loss of VAP association with the inclusion, respectively (*Figure 6B, C*) indicating that substitution of $T_{265}$ to a nonphosphorylatable residue impacted the ability of the noncanonical FFAT motif to mediate the VAP–FFAT interaction. On the contrary, IncV$_{T265D/P266A}$ resulted in VAP inclusion association similar to IncV$_{WT}$, indicating that phosphomimetic mutation of IncV noncanonical FFAT resulted in optimal VAP binding during infection. However, IncV$_{T265D/P266A/Y287A}$ did not result in the expected intermediate VAP recruitment observed with IncV$_{Y287A}$, suggesting that the T265D/P266A mutation of the noncanonical FFAT was not sufficient to rescue VAP binding in the background of an inactivated canonical FFAT.

We next investigated if phosphomimetic mutation of $T_{265}$ of the IncV noncanonical FFAT (IncV$_{T265D/P266A}$) was sufficient to promote the IncV–VAP interaction in vitro. GST-IncV$_{T265D/P266A}$ failed to interact with MBP-VAP$_{MSP}$ in our in vitro assay (*Figure 6—figure supplement 2*), suggesting that phosphorylation of $T_{265}$ is not sufficient for full VAP binding under these conditions.

To further investigate if the noncanonical FFAT of IncV could function as a phospho-FFAT motif, we used a VAP$_{K50L}$ mutant. The K50L mutation resides in the MSP domain of VAP and has been shown to specifically affect VAP binding to phospho-FFAT motifs, but not to canonical FFAT (*Di Mattia et al., 2020*). Using a similar experimental setup as in *Figure 6B, C*, we investigated the ability of YFP-VAP$_{K50L}$ to associate with inclusions harboring bacteria expressing IncV$_{WT}$-, IncV$_{Y287A}$-, or IncV$_{T265A}$-3xFLAG (*Figure 6—figure supplement 3*). Cells expressing YFP-VAP$_{WT}$ and infected with bacteria expressing IncV$_{WT}$- or IncV$_{F263A/Y287A}$-3xFLAG were used as positive and negative controls, respectively. In comparison to these controls, IncV$_{WT}$ led to an intermediate recruitment of VAP$_{K50L}$ to the inclusion, presumably due to binding to the canonical FFAT only, which was supported by the inability of IncV$_{Y287A}$ to recruit VAP$_{K50L}$. IncV$_{T265A}$ also led to an intermediate recruitment of VAP$_{K50L}$ to the inclusion, which was attributed to binding to the canonical FFAT. However, IncV$_{T265A}$ displayed decreased efficiency in VAP recruitment compared to IncV$_{WT}$, suggesting that in the context of IncV, VAP$_{K50L}$ may have low residual affinity to the noncanonical FFAT containing a phosphomimetic mutation of $T_{265}$, and/or reduced activity toward the canonical FFAT.

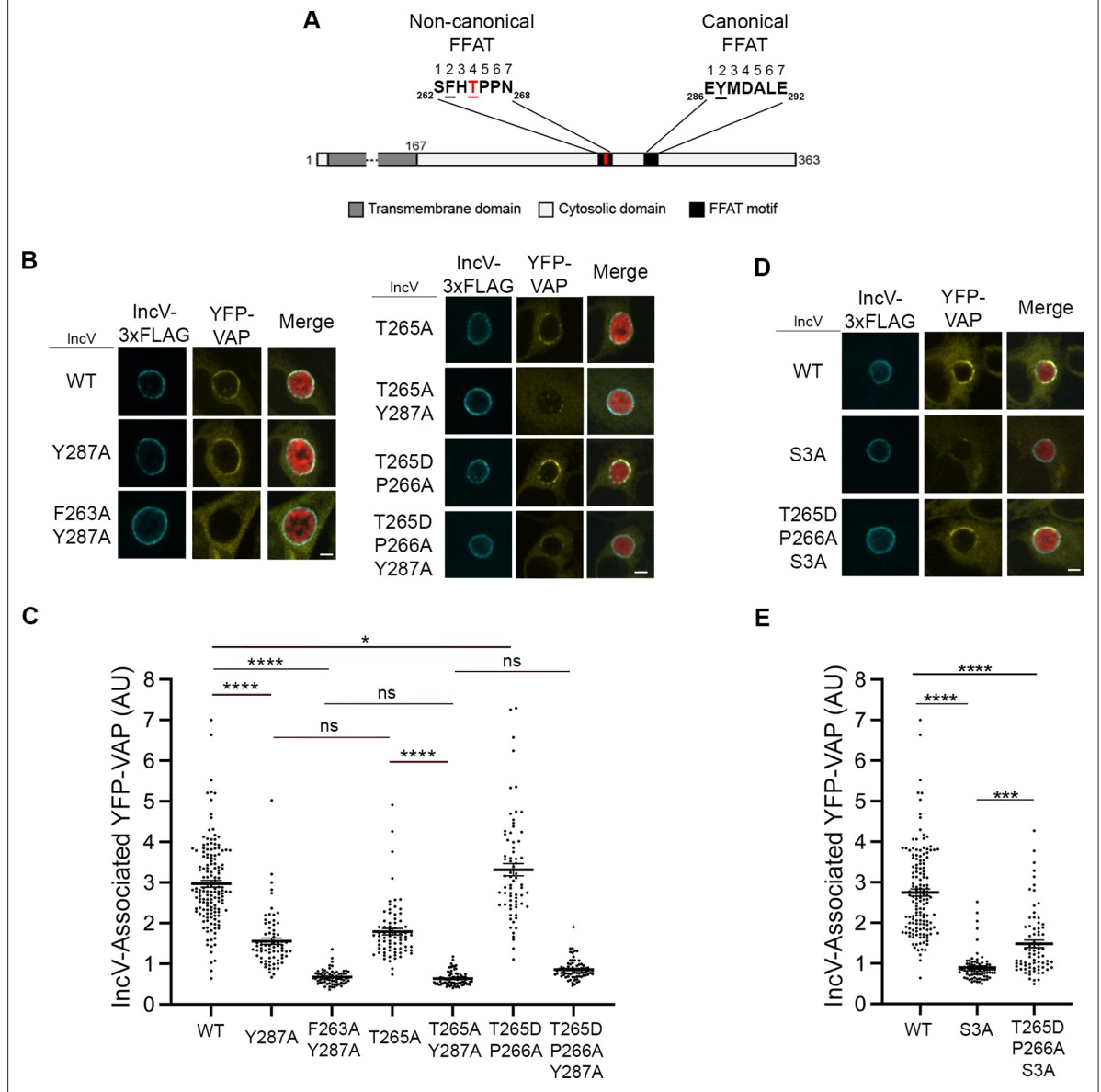

**Figure 6.** Phosphorylation of T$_{265}$ in the noncanonical FFAT motif of IncV contributes to the IncV–VAP interaction. (**A**) Schematic depicting the IncV protein. The transmembrane domain, the cytosolic domain, and the noncanonical and the canonical FFAT motif cores are indicated in dark gray, light gray, and black, respectively. The amino acid sequence of the FFAT motif cores is shown. Numbers 1–7 indicate the amino acid position within the FFAT motif cores, other numbers indicate the amino acid position within the IncV protein sequence. Residues at position 2 of the FFAT motif cores are in black and underlined. Threonine 265 at position 4 of the noncanonical FFAT is in red and underlined. (**B**) Single plane confocal images of HeLa cells expressing YFP-VAP (yellow), infected with a *C. trachomatis incV* mutant expressing mCherry constitutively (red) and IncV$_{WT}$- (WT), IncV$_{Y287A}$- (Y287A), IncV$_{F263A/Y287A}$ (F263A/Y287A), IncV$_{T265A}$- (T265A), IncV$_{T265A/Y287A}$- (T265A/Y287A), IncV$_{T265D/P266A}$- (T265D/P266A), or IncV$_{T265D/P266A/Y287A}$-3xFLAG (T265D/P266A/Y287A) (blue). The merge is shown on the right. Scale bar is 5 µm. (**C, E**) Quantification of the mean intensity of YFP-VAP within an object generated from the IncV-3xFLAG signal corresponding to the indicated IncV constructs, and normalized to the mean intensity of YFP-VAP in the ER. Each dot represents one inclusion. Data show the mean and SEM of three to six independent experiments. One-way ANOVA and Tukey's post hoc test were performed. ns: nonsignificant, *p < 0.05, ***p < 0.001, ****p < 0.0001. (**D**) Single plane confocal images of HeLa cells expressing YFP-VAP (yellow), infected with a *C. trachomatis incV* mutant expressing mCherry constitutively (red) and IncV$_{WT}$- (WT), IncV$_{S3A}$- (S3A), or IncV$_{T265D/P266A/S3A}$-3xFLAG (T265D/P266A/S3A) (blue). The merge is shown on the right. Scale bar is 5 µm.

The online version of this article includes the following source data and figure supplement(s) for figure 6:

**Source data 1.** Quantification of IncV-associated YFP-VAP for *Figure 6*.

**Figure supplement 1.** Inclusion localization of IncV variants with amino acid substitutions in the canonical and/or noncanonical FFAT motifs.

**Figure supplement 1—source data 1.** Quantification of inclusion-associated IncV for *Figure 6—figure supplement 1*.

*Figure 6 continued on next page*

*Figure 6 continued*

**Figure supplement 2.** Phosphomimetic mutation of $T_{265}$ in the IncV noncanonical FFAT is not sufficient to promote the IncV–VAP interaction in vitro.

**Figure supplement 2—source data 1.** Quantification of blot densities for *Figure 6—figure supplement 2B*.

**Figure supplement 2—source data 2.** Uncropped, labeled blots for *Figure 6—figure supplement 2B*.

**Figure supplement 2—source data 3.** Raw data for MBP blot in *Figure 6—figure supplement 2B*.

**Figure supplement 2—source data 4.** Raw data for Ponceau S blot in *Figure 6—figure supplement 2B*.

**Figure supplement 3.** Recruitment of $VAP_{WT}$ and $VAP_{K50L}$ to *C. trachomatis* inclusions displaying wild-type or mutated IncV.

**Figure supplement 3—source data 1.** Quantification of IncV-associated YFP-VAP for *Figure 6—figure supplement 3*.

Although $T_{265}$ cannot be phosphorylated by CK2, given the CK2-dependent role of $S_{345}$, $S_{346}$ and $S_{350}$ in controlling the global level of phosphorylation of IncV, we investigated if CK2 could indirectly affect phosphorylation of $T_{265}$, by testing the ability of the $T_{265}$ phosphomimetic mutation to bypass the need for CK2 for VAP recruitment to the inclusion. We generated an $IncV_{T265D/P266A/S3A}$ construct and quantified VAP recruitment to inclusions harboring bacteria expressing $IncV_{T265D/P266A/S3A}$-3xFLAG (*Figure 6D, E*). $IncV_{WT}$- and $IncV_{S3A}$-3xFLAG were used as positive and negative controls, respectively. $IncV_{T265D/P266A/S3A}$ inclusion localization was not affected (*Figure 6—figure supplement 1B*) and led to a partial rescue of VAP recruitment to the inclusion, compared to the controls (*Figure 6D, E*). These results indicate that CK2 and $S_{345}$, $S_{346}$, and $S_{350}$ play an indirect role in the phosphorylation of $T_{265}$ in the IncV noncanonical FFAT motif.

Altogether, because of inconclusive results with the $VAP_{K50L}$ mutant, it remains unclear if IncV noncanonical FFAT acts as a phospho-FFAT. However, the above results indicate that phosphorylation of $T_{265}$ of IncV noncanonical FFAT is indirectly controlled by CK2 and contributes, but is not sufficient, to mediate the IncV–VAP interaction, suggesting that additional residues must be phosphorylated to ensure full IncV–VAP interaction.

## Phosphorylation of serine-rich tracts upstream of IncV FFAT motifs substitute typical acidic tracts and are key for the IncV–VAP interaction

In addition to the seven amino acid core of the FFAT motif, VAP–FFAT-mediated interactions also rely on the presence of acidic residues upstream of the core sequence, referred to as the acidic tract. It allows for the initial electrostatic interaction with VAP by interacting with the electropositive charge of the MSP domain before the FFAT core motif locks into its dedicated groove (*Furuita et al., 2010*). We noted that, instead of typical acidic residues, the primary amino acid structures upstream of the IncV FFAT motifs are highly enriched in phosphorylatable serine residues (*Figure 7A*). We hypothesized that, if phosphorylated, these serine residues could serve as an acidic tract and facilitate the IncV–VAP interaction. To test this hypothesis, the 10 residues directly upstream of the noncanonical FFAT motif and the eight residues directly upstream of the canonical FFAT motif were mutated to alanine residues (referred to as $IncV_{S/A}$). The ability of $IncV_{S/A}$-3xFLAG to recruit VAP to the inclusion was assessed. HeLa cells expressing YFP-VAP were infected with *C. trachomatis incv* mutant strains expressing either $IncV_{WT}$-, $IncV_{F263A/Y287A}$-, or $IncV_{S/A}$-3xFLAG under an aTc-inducible promoter. The cells were fixed at 24 h pi and analyzed by confocal immunofluorescence microscopy (*Figure 7B*). All IncV constructs were equally localized to the inclusion membrane (*Figure 7—figure supplement 1*). Qualitative and quantitative analysis revealed that expression of $IncV_{S/A}$-3xFLAG resulted in a significant decrease in YFP-VAP recruitment to the inclusion as observed with $IncV_{F263A/Y287A}$-3xFLAG and compared to $IncV_{WT}$-3xFLAG (*Figure 7B, C*). To determine if this decrease in VAP recruitment was due to a lack of CK2 recruitment, the ability of these strains to recruit YFP-CK2β to the inclusion was assessed by confocal microscopy (*Figure 7—figure supplement 2*). All three strains recruited CK2 to the inclusion, indicating that the lack of VAP recruitment upon expression of $IncV_{S/A}$ was not due to a lack of CK2 recruitment.

We next determined if phosphomimetic mutation of the serine-rich tracts of IncV to aspartic acid residues (referred to as $IncV_{S/D}$) was sufficient to rescue the ability of the cytosolic domain of IncV expressed in *E. coli* to interact with the MSP domain of VAP in our VAP binding in vitro assay. As observed before, there was minimal binding of $VAP_{MSP}$ to $IncV_{WT}$ (*Figure 7D*, lane 2). However, we observed a 20-fold increase in $VAP_{MSP}$ binding to $IncV_{S/D}$ compared to $IncV_{WT}$ (*Figure 7D*, lane 3), indicating that phosphomimetic mutation of the serine-rich tracts is sufficient to promote the IncV–VAP interaction in vitro. Altogether, these results indicate that instead of typical acidic tracts,

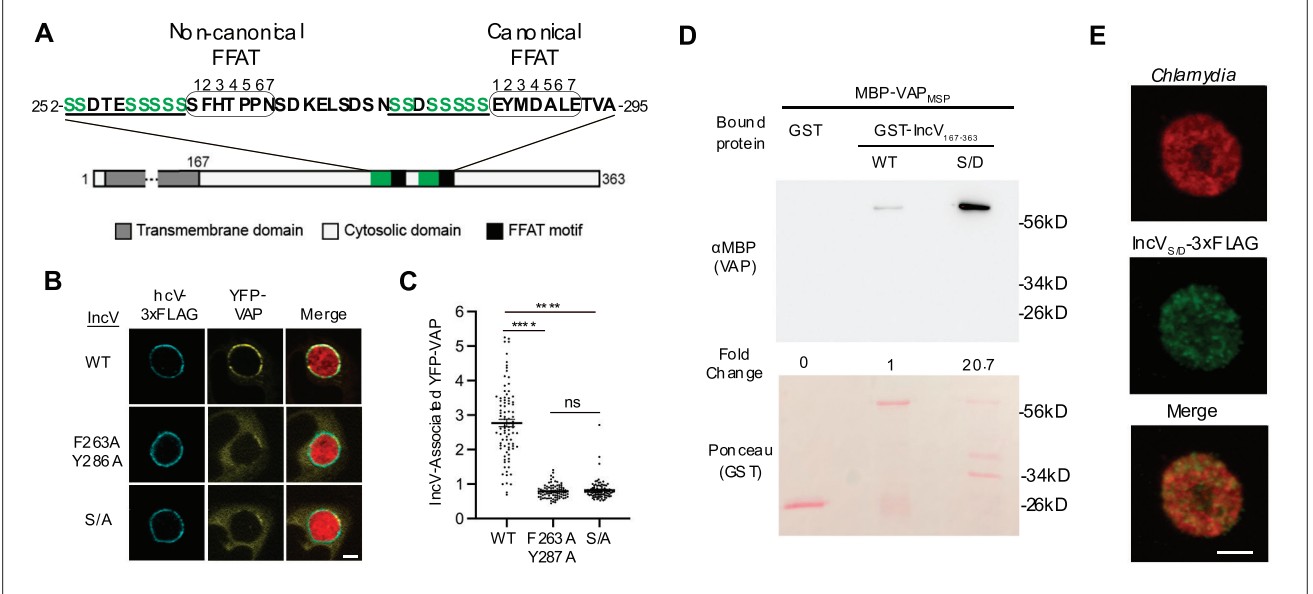

**Figure 7.** Phosphorylation of the serine tracts upstream of the IncV FFAT motifs facilitates the IncV–VAP interaction. (**A**) Schematic depicting the IncV protein. The transmembrane domain, the cytosolic domain, and the noncanonical and canonical FFAT motif cores are indicated in dark gray, light gray, and black, respectively. The amino acid sequence of the FFAT motif cores (circled) and their respective upstream sequence is shown. The serine-rich tracts are underlined. Serine residues are in green. Numbers 1–7 indicate the amino acid position within the FFAT motif cores, other numbers indicate the amino acid position within the IncV protein sequence. (**B**) Single plane confocal images of HeLa cells expressing YFP-VAP (yellow), infected with a *C. trachomatis incV* mutant expressing mCherry constitutively (red) and IncV$_{WT}$-3xFLAG (WT), IncV$_{F263A/Y287A}$-3xFLAG (F263A/Y287A), or IncV$_{S/A}$-3xFLAG (S/A) (blue). The merge is shown on the right. Scale bar is 5 µm. (**C**) Quantification of the mean intensity of YFP-VAP within an object generated from the IncV-3xFLAG signal and normalized to the mean intensity of YFP-VAP in the ER. Each dot represents one inclusion. Data show the mean and SEM of a combination of three independent experiments. One-way ANOVA with Tukey's post hoc test was performed. ****$p < 0.0001$. (**D**) In vitro binding assay using GST, GST-IncV$_{WT}$, or GST-IncV$_{S/D}$ purified from *E. coli*, and immobilized on glutathione beads and combined with MBP-VAP purified from *E. coli*. The top panel was probed with anti-MBP and the bottom panel was the same membrane stained with Ponceau S to detect the GST construct. Note that the IncV and VAP constructs, only include the cytosolic domain of IncV (aa 167–363) and the MSP domain of VAP, respectively. (**E**) Single plane confocal images of HeLa cells infected with a *C. trachomatis incV* mutant expressing mCherry consitutively (red) and IncV$_{S/D}$-3xFLAG (green). The merge is shown on the bottom. Scale bar is 5 µm.

The online version of this article includes the following source data and figure supplement(s) for figure 7:

**Source data 1.** Quantification of IncV-associated YFP-VAP and blot densities for *Figure 7*.

**Source data 2.** Uncropped, labeled blots for *Figure 7D*.

**Source data 3.** Raw data for MBP blot in *Figure 7D*.

**Source data 4.** Raw data for Ponceau S blot in *Figure 7D*.

**Figure supplement 1.** Alanine substitution of IncV serine tracts does not affect IncV inclusion localization.

**Figure supplement 1—source data 1.** Quantification of inclusion-associated IncV for *Figure 7—figure supplement 1*.

**Figure supplement 2.** Alanine substitution of residues in position 2 of IncV FFAT motifs or of the serine-rich tracts upstream of IncV FFAT motifs does not affect IncV-dependent CK2 recruitment to the inclusion.

**Figure supplement 2—source data 1.** Quantification of IncV-associated YFP-CK2β for *Figure 7—figure supplement 2*.

phosphorylated serine-rich tracts located upstream of IncV FFAT motifs are both necessary and sufficient for promoting the IncV–VAP interaction.

In order to confirm the role of IncV serine-rich tracts in promoting the IncV–VAP interaction during infection, we assessed the ability of IncV$_{S/D}$-3xFLAG to recruit VAP to the inclusion when expressed from an *incV* mutant strain of *C. trachomatis*. In comparison to IncV$_{WT}$ and all other mutated alleles used in this study, IncV$_{S/D}$-3xFLAG remained trapped within the bacteria and did not localize to the inclusion membrane (*Figure 7E*). These results suggest that phosphorylatable serine residues may have been selected over acidic residues to allow proper Type III translocation of IncV to the inclusion membrane.

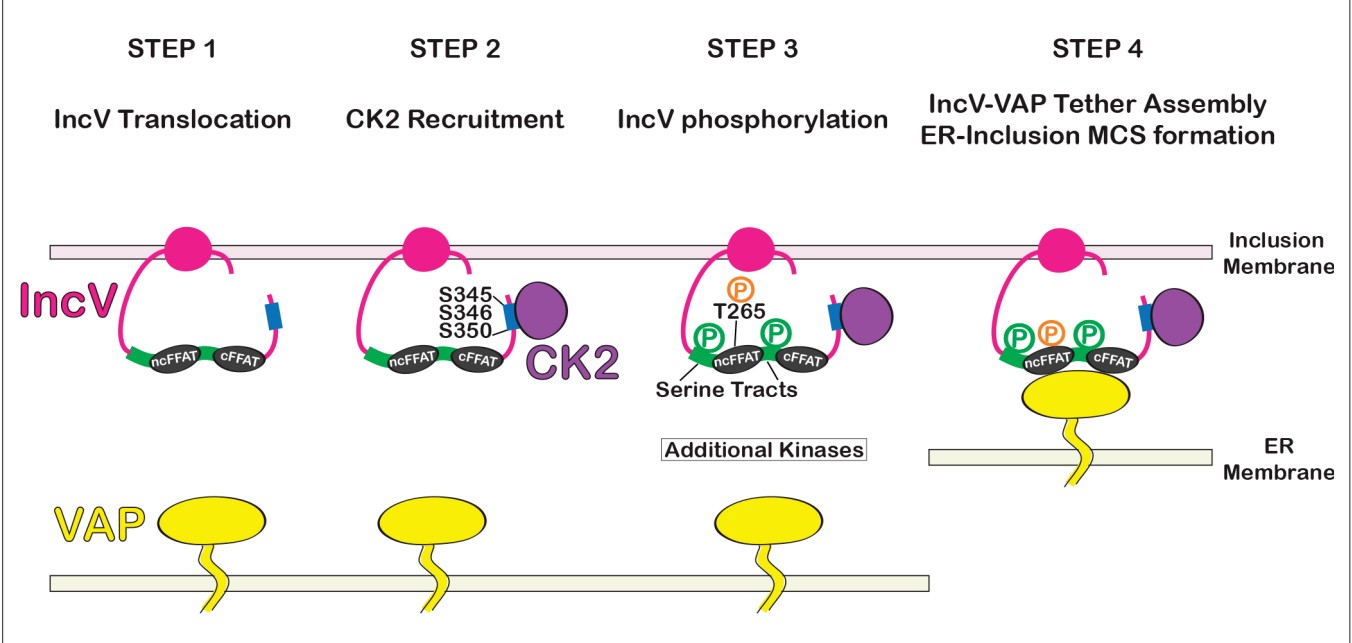

**Figure 8.** Model of assembly of the IncV–VAP tether at ER-Inclusion membrane contact site (MCS). Step 1: After Type III-mediated translocation, unphosphorylated IncV (pink) is inserted into the inclusion membrane. Step 2: IncV recruits CK2 (purple) via three serine residues S345, S346, and S350 (blue) that are part of CK2 recognition motifs and located in a C-terminal domain of IncV. Step 3: IncV becomes hyperphosphorylated, including phosphorylation of the noncanonical FFAT on threonine residue T265 (orange) and the serine-rich tract (green) immediately upstream of FFAT core motifs (black). Additional kinases contribute to IncV phosphorylation. Step 4: IncV phosphorylation leads to full mimicry of FFAT motifs and binding to VAP (yellow) resulting in ER-Inclusion MCS formation. P: phosphorylated residues; ncFFAT: noncanonical FFAT; cFFAT: canonical FFAT. Light pink: inclusion membrane. Light yellow: ER membrane.

## Discussion
### IncV/CK2/VAP interplay in ER-Inclusion MCS formation

Based on our results, we propose the following model of assembly of the IncV–VAP tether leading to ER-Inclusion MCS formation during *Chlamydia* infection (*Figure 8*). Unphosphorylated IncV is translocated across the inclusion membrane by the T3SS. Upon insertion into the inclusion membrane and exposure to the cytosol (*Figure 8*, Step 1), IncV recruits the host kinase CK2 through a C-terminal domain containing three serine residues ($S_{345}$, $S_{346}$, and $S_{350}$) that are part of CK2 recognition sites (*Figure 8*, Step 2). As a consequence, IncV becomes hyperphosphorylated, including phosphorylation of $T_{265}$ of the noncanonical FFAT and serine tracts directly upstream of the FFAT motifs (*Figure 8*, Step 3). These phosphorylation events lead to IncV interaction with VAP, tether assembly, and ER-Inclusion MCS formation (*Figure 8*, Step 4). Our data are consistent with CK2 playing a central role in initiating IncV hyperphosphorylation (*Figure 8*, Step 2), however, additional kinases must contribute, especially for phosphorylation of $T_{265}$ (*Figure 8*, Step 3). Importantly, the post-translocation phosphorylation of IncV ensures optimal VAP binding while preserving proper T3SS-mediated translocation of IncV to the inclusion membrane. Although this model was in part inferred from overexpression studies, it constitutes a framework to guide futures studies to further dissect the complex phosphorylation events leading to the assembly of the endogenous IncV–VAP tether. Below we discuss our results in the context of emerging regulatory mechanisms of cellular MCS assembly and highlight conserved and pathogen-specific mechanisms.

### IncV-dependent recruitment of CK2 to the inclusion

Few kinases phosphorylating VAP-dependent tethers have been identified (*Kors et al., 2022*; *Xu et al., 2020*) and how they associate with MCS to phosphorylate their target has not been explored. Here, we show that IncV recruits CK2 to ER-Inclusion MCS through interaction with its C-terminal domain, a mandatory step for IncV hyperphosphorylation. Three serine residues ($S_{345}$, $S_{346}$, and $S_{350}$)

that match the CK2 recognition motifs (S-x-x-D/E) located in a C-terminal domain of IncV are essential for CK2 recruitment to the inclusion, and IncV phosphorylation.

However, while CK2 was required for ER-Inclusion MCS formation during infection, and sufficient for IncV–VAP interaction in vitro, the introduction of phosphomimetic mutations at $S_{345}$, $S_{346}$, and $S_{350}$ was not sufficient to promote the IncV–VAP interaction in vitro. While it is possible that the phospho-mimetic mutations failed to mimic phosphorylation, it is more likely that additional phosphorylation sites exist. Kinase-substrate recognition is a complex process that goes beyond the simple recognition of a consensus sequence and can involve docking sites away from the phosphorylation sites (*Miller and Turk, 2018*). The cytosolic domain of IncV contains a large number of additional potential CK2 recognition sites. Among those, two serine residues, $S_{253}$ and $S_{283}$, located in the serine tracts upstream of the noncanonical FFAT and the canonical FFAT, respectively, are direct CK2 targets. Additionally, serine residues $S_{257}$, $S_{258}$, and $S_{259}$ (serine tract upstream of the noncanonical FFAT), and $S_{278}$, $S_{281}$, $S_{282}$, and $S_{284}$ (serine tract upstream of the canonical FFAT) can be phosphorylated upon priming the serine or tyrosine residue at the last position of the CK2 recognition motif (S/T-x-x-D/E/pS/pY). We therefore propose that alanine substitution of $S_{345}$, $S_{346}$, and $S_{350}$ eliminates an essential docking site for subsequent CK2-mediated phosphorylation of distal residues in the cytosolic domain of IncV, including the serine tracts next to the FFAT motifs (see below). Further investigation of the IncV-dependent recruitment of CK2 to ER-Inclusion MCS could offer some insights into kinase targeting to cellular MCS. Moreover, since intracellular pathogens often mimic cellular processes, our study may have identified CK2 as a regulator of cellular MCS.

## IncV noncanonical FFAT motif and the role of $T_{265}$

Phospho-null mutation of $T_{265}$ was detrimental to the IncV-dependent recruitment of VAP to the inclusion. Conversely, phosphomimetic mutation of $T_{265}$, in the context of an intact canonical-FFAT, resulted in efficient VAP association with the inclusion. These results are similar to those obtained with the phospho-FFAT motif-containing proteins STARD3 and VPS13D (*Di Mattia et al., 2020*; *Guillén-Samander et al., 2021*), and could therefore, suggest that, as previously proposed by Di Mattia et al., the noncanonical FFAT of IncV, via phosphorylation of $T_{265}$, is acting as a phospho-FFAT to mediate direct IncV–VAP interaction. However, in the absence of phospho-proteomic data demonstrating phosphorylation of $T_{265}$, and inconclusive and discrepant results discussed below, further investigation is required to validate that IncV noncanonical FFAT is a phospho-FFAT. First, in the context of a mutated canonical FFAT motif, phosphomimetic of $T_{265}$ was unable to confer the anticipated partial rescue of VAP recruitment to the inclusion. The T265D/P266A phosphomimetic mutation was based on STARD3, where the proline residue directly downstream of the threonine residue had to be mutated to an alanine to introduce flexibility in the amino acid chain and allow for the aspartic acid to act as a phosphomimetic (*Di Mattia et al., 2020*). This approach was successful to rescue VAP binding to VAP interacting proteins containing only a single phospho-FFAT motif, like STARD3 and VPS13D (*Di Mattia et al., 2020*; *Guillén-Samander et al., 2021*), and to IncV containing an additional wild-type canonical FFAT, but it may not be optimal for IncV in the context of a mutated canonical FFAT motif, perhaps because of the tertiary structure of the protein. Second, in the context of IncV, the $VAP_{K50L}$ mutation led to partially inconclusive results, which could be due to low residual binding of $VAP_{K50L}$ to IncV noncanonical FFAT motif and/or an overall reduced activity toward the canonical FFAT. Further characterization of the $VAP_{K50L}$ mutant, especially in the context of proteins containing both a phospho-FFAT and a canonical FFAT, could settle these possibilities. Third, in vitro, the phosphomimetic mutation of $T_{265}$ was not sufficient to promote VAP binding, suggesting that phosphorylation of $T_{265}$ is not required in this setting. This conclusion is in line with the fact that, in vitro, CK2 phosphorylated IncV interacts with VAP, despite the fact that $T_{265}$ is not a CK2 target and is therefore presumably not phosphorylated in this particular experiment. Collectively, these results suggest that in vitro CK2 phosphorylation of IncV may result in suboptimal binding to VAP, which could be further increased during infection by phosphorylation of $T_{265}$. It is also possible that the dynamics of IncV–VAP interaction are different in vitro, compared to infection conditions, and that phosphorylation of $T_{265}$ is not needed in this context. Alternatively, it is possible that phosphorylation of $T_{265}$ is necessary in vivo for something other than VAP binding, such as the recruitment or activation of CK2, a step that may be bypassed in vitro given the high amount of CK2. Similarly, the positive in vitro interaction between $IncV_{S/D}$ and VAP, in the absence of $T_{265}$ phosphorylation, could indicate either suboptimal in vitro binding that could

be strengthened in vivo by phosphorylation of $T_{265}$, or a function independent of VAP binding and leading to phosphorylation of the serine tracts. The latter would be consistent with the partial bypass of the need for CK2 for VAP association with the inclusion upon phosphomimetic mutation of $T_{265}$. This result also suggests that phosphorylation of $T_{265}$ is dependent on, and temporally downstream of, CK2 binding to $S_{345}$, $S_{346}$, and $S_{350}$ in the C-terminal of IncV, potentially by allowing the recruitment of the yet to be identified kinase responsible for phosphorylating $T_{265}$.

The presence of a phospho-FFAT in IncV would add to the growing list of proteins that interact with VAP via a phospho-FFAT. These include STARD3 at ER–endosome contacts, the potassium channel Kv.2 at ER–PM contacts in neurons, Miga at ERMCS, and VPS13D at ER–mitochondria MCS (*Di Mattia et al., 2020*; *Guillén-Samander et al., 2021*; *Johnson et al., 2018*; *Xu et al., 2020*). Moreover, although not recognized as such at the time, a putative phospho-FFAT in the norovirus protein NS2 is essential for interaction with VAP and viral replication (*McCune et al., 2017*), indicating that this mechanism of interaction with VAP might also be conserved amongst pathogens.

In the context of the STARD3-dependent formation of ER–endosome contacts, the presence of a single phospho-FFAT is proposed to act as a molecular switch to regulate contact formation (*Di Mattia et al., 2020*). In the case of proteins that contains a combination of a canonical FFAT and a phospho-FFAT, such as PTPIP51, an ER–mitochondria contact protein (*Di Mattia et al., 2020*) and potentially IncV, it is unclear how a most likely constitutive canonical FFAT and a regulated FFAT motif cooperate, if one is dominant over the other, and how advantageous such a combination is with respect with MCS regulation. In the case of IncV, one could speculate that the canonical FFAT motif allows for a baseline level of VAP recruitment to the inclusion and MCS formation while a phospho-FFAT could increase VAP recruitment beyond this baseline.

## The IncV–VAP interaction is mediated by phosphorylatable serine tracts

In eukaryotic FFAT motifs, a number of negative charges upstream of the FFAT motif are proposed to facilitate the initial interaction with the MSP domain of VAP (*Furuita et al., 2010*). This negatively charged surface is conferred by acidic residues, but phosphorylated residues have been implicated in a few instances. The phosphorylation, by an unknown kinase, of a single serine residue six amino acids upstream of the CERT FFAT motif ($S_{315}$) enhances the CERT–VAP interaction (*Kumagai et al., 2014*). Similarly, the phosphorylation of three serine residues directly upstream of the FFAT-like motif of ACBD5 promotes VAP binding and peroxisome–ER contacts formation (*Kors et al., 2022*). We note that two of these serine residues ($S_{259}$ and $S_{261}$) are putative CK2 targets, so it would be interesting to determine if CK2 also plays a role in assembly of the ACBD5-VAP tether. Another example are six serine residues, spread over 21 residues upstream of the core FFAT motif of Miga, shown to facilitate the Miga–VAP interaction (*Xu et al., 2020*). At least two kinases CKI and CaMKII were required for Miga phosphorylation; however, other kinases are likely involved (*Xu et al., 2020*). In the case of IncV, the mimicry of an acidic track via phosphorylatable residues seems to be brought to the extreme, since the eight to ten amino acid stretch directly preceding each FFAT motif include 80–87% of serine residues, the remaining residues being acidic. Except for OSBP2/ORP4, which contains six acidic residues (including a phosphorylatable threonine), most acidic tracts contain few acidic residues directly upstream of the core of the FFAT motif (*Neefjes and Cabukusta, 2021*). IncV is the first example of a FFAT motif-containing protein that displays serine tracts in place of acidic tracts. If built into the available FFAT motif identification algorithms, this feature could potentially reveal additional cellular VAP interacting proteins (*Di Mattia et al., 2020*; *Murphy and Levine, 2016*).

## Phosphoregulation and pathogenesis

During coevolution with the mammalian host, obligate intracellular bacteria such as *Chlamydia* have evolved to take advantage of and manipulate cellular machinery. One mechanism is via molecular mimicry, in which the pathogen mimics features that are uniquely present in host proteins (*Mondino et al., 2020*). In the case of IncV and acidic tracks in FFAT motifs, however, one could wonder why evolution would converge toward a mechanism relying on phosphorylation by host cell kinases, as opposed to simply selecting for genetically encoded acidic residues. In the case of *Chlamydia* Inc proteins, it is possible that tracks of aspartic acid or glutamic acid residues would create an excess of negative charges that may interfere with Type III secretion. In support of this notion, we found that

*Chlamydia* IncV is no longer properly translocated to the inclusion membrane when the serine tracts are mutated to aspartic acid residues, and instead remains trapped within the bacteria. Our results support the notion that the recruitment of CK2 to the inclusion supports the assembly of the IncV–VAP tether. In addition, we cannot exclude the possibility that the recruitment of a phosphatase to ER-Inclusion MCS may contribute to the disassembly of IncV–VAP tethers, as shown for the calcineurin-dependent disassembly of Kv.2-VAP ER–PM contacts in neurons (*Park et al., 2006*). A combination of host cell kinases and phosphatases could thus regulate the dynamics of ER-Inclusion contact sites during the *Chlamydia* developmental cycle.

## Materials and methods

### Cell lines and bacterial strains

HeLa cells (ATCC CCL-2) and HEK293 cells (ATCC CRL-1573) were maintained in DMEM high glucose (Gibco) containing 10% heat-inactivated fetal bovine serum (Gibco) at 37°C and 5% $CO_2$. *C. trachomatis* Lymphogranuloma venereum, Type II (ATCC L2/434/Bu VR-902B) was propagated in HeLa cells as previously described (*Derré et al., 2007*). The *incV::bla* mutant strain of *C. trachomatis* (also known as *CT005::bla*) was obtained from Ted Hackstadt (NIH, Rocky Mountain Laboratories) (*Weber et al., 2017*). All cell lines and *Chlamydia* strains are routinely tested for mycoplasma contamination.

### Plasmid construction

Inserts were generated by PCR using the primers (IDT) and templates listed in *Supplementary file 1*, and the Herculase DNA polymerase (Stratagene). The inserts were cloned as described below using restriction enzymes (NEB) and T4 DNA ligase (NEB).

### Vectors for expression in mammalian cells

The IncV-3xFLAG construct cloned in the pCMV-IE-N2-3xFLAG vector was previously described (*Stanhope et al., 2017*). The YFP-CK2α and YFP-CK2β plasmids were kind gifts from Claude Cochet and Odile Filhol-Cochet (Institut Albert Bonniot Departement Reponse et Dy-namique Cellulaires) and were previously characterized (*Filhol et al., 2003*). The pCFP-VAP and pYFP-VAP plasmids were constructed by cloning the VAPA open reading frame (ORF) into pCMV-N1-CFP and pCMV-N1-YFP, respectively, using AgeI and HindIII restriction sites. The pYFP-VAP$_{K50L}$ plasmid was constructed by cloning the VAPA ORF encoding the K50L mutation (generated my overlapping PCR), into pCMV-N1-YFP using AgeI and HindIII restriction sites.

### Vectors for expression in *E. coli*

The GST-VAP$_{MSP}$ plasmid was previously described (*Stanhope et al., 2017*). MBP-VAP$_{MSP}$ was constructed by cloning the MSP domain of VAPA using NotI and BamHI into pMAL. The GST-IncV$_{167-363}$ fusion constructs WT, S3D, S/D, or T265D/P266A were generated by cloning a DNA fragment encoding amino acids 167–363 of IncV with the indicated mutations into the BamHI and XhoI restriction sites of pGEX-KG.

### Vectors for expression in *C. trachomatis*

All the constructs to express the various IncV-3xFLAG constructs (1–356, 1–341, S3A, T265A, T265A/Y287A, T265D/P266A, T265D/P266A/Y287A, T265D/P266A/S3A, S/A, and S/D) from the aTc-inducible promoter were generated by overlapping PCR and cloned into KpnI and NotI of the p2TK2$_{Spec}$-SW2 mCh(Gro) vector (*Cortina et al., 2019*). The Tet-IncV$_{FL}$, Tet-IncV$_{F263A/Y287A}$, and Tet-IncV$_{Y287A}$ constructs were previously described (*Stanhope et al., 2017*).

### *C. trachomatis* transformation and *incV::bla* complementation

Wild-type *C. trachomatis* or an *incV* mutant (*incV::bla*) were transformed using our previously described calcium-based *Chlamydia* transformation procedure (*Cortina et al., 2019*).

### DNA transfection

Cells were transfected with mammalian construct DNA according to the manufacturer's instructions with X-tremeGENE 9 DNA Transfection Reagent (Roche).

## Sodium dodecyl sulfate–polyacrylamide gel electrophoresis

Cells were either directly lysed in 2× Laemmli buffer with 10 mM DTT(Dithiothreitol) or IncV was purified as described in the immunoprecipitation and protein purification sections then suspended in a final concentration of 1× Laemmli buffer with 10 mM DTT. Protein samples were separated using Sodium dodecyl sulfate–polyacrylamide gel electrophoresis (SDS–PAGE).

## Immunoblotting

After SDS/PAGE, proteins were transferred onto nitrocellulose membranes (GE Healthsciences). Prior to blocking, membranes were stained with Ponceau S in 5% acetic acid and washed in $dH_2O$. Membranes were incubated for 1 hr with shaking at room temperature in blocking buffer (5% skim milk in 1× phosphate-buffered saline [PBS] with 0.05% Tween). Membranes were then incubated with primary and secondary (horseradish peroxidase [HRP]-conjugated) antibodies diluted in blocking buffer overnight at 4°C and 1 hr at room temperature, respectively, with shaking. ECL Standard western blotting detection reagents (Amersham) were used to detect HRP-conjugated secondary antibodies on a BioRad ChemiDoc imaging system. CK2β was detected using secondary antibodies conjugated to Alexa Fluor 800 on Li-Cor Odyssey imaging system.

## Antibodies

The following antibodies were used for immunofluorescence microscopy (IF) and immunoblotting (WB): mouse monoclonal anti-FLAG [1:1000 (IF); 1:10,000 (WB); Sigma], rabbit polyclonal anti-CK2β [1:200 (IF); 1:1000 (WB); Bethyl Antibodies]; rabbit polyclonal anti-thiophosphate ester antibody [1:2000 (WB); Abcam], rabbit polyclonal anti-MBP [1:10,000 (WB); NEB], rabbit polyclonal anti-GAPDH [1:10,000 (WB)], rabbit polyclonal anti-mCherry [1:2,000 (WB); BioVision], rabbit polyclonal anti-actin [1:10,000 (WB); Sigma], HRP-conjugated goat anti-rabbit IgG [1:10,000 (WB); Jackson], HRP-conjugated goat anti-mouse IgG [1:10,000 (WB); Jackson], Alexa Fluor 514-, 800-, or Pacific Blue-conjugated goat anti-mouse IgG [1:500 (IF); 1:10,000 (WB); Molecular Probes].

## Immunoprecipitation of IncV-3xFLAG from HEK293 cells infected with *C. trachomatis*

800,000 HEK293 cells were seeded into one well of a 6-well plate (Falcon) and infected the following day with *C. trachomatis* at a multiplicity of infection (MOI) of 5. 8 h pi, media containing 2 ng/ml aTc was added to the infected cells for 16 hr. Twenty-four hours postinfection, culture media was removed from the cells and 500 µl of lysis buffer (20 mM Tris pH 7.5, 150 mM NaCl, 2 mM EDTA (Ethylenedi-aminetetraacetic acid), 1% Triton X-100, protease inhibitor mixture EDTA-free [Roche]) was added per well. Cells were lysed for 20 min at 4°C with rotation. Lysates were centrifuged at 16,000 × *g* for 10 min at 4°C to pellet nuclei and unlysed cells. Cleared lysates were incubated with 10 µl of anti-FLAG M2 affinity beads (Sigma) for 2 hr at 4°C with rotation. The beads were washed with lysis buffer three times. Proteins were eluted with 50 µl of 100 µg/ml 3xFLAG peptide (Sigma) in 1× Tris-buffered saline (TBS). For cells transfected with pCMV-IE-N2-IncV-3xFLAG, cells were not infected, and the remainder of the protocol remained the same starting with removal of media and lysing.

## Phosphatase assay

Immunoprecipitation was performed as described above with the following changes: The beads were washed with 1× TBS three times and proteins were eluted with 55 µl of 100 µg/ml 3xFLAG peptide (Sigma) in 1× TBS. 20 µl of eluate was combined with 2.5 µl of 10 mM $MnCl_2$, 2.5 µl of 10× PMP buffer (NEB), and 400 units of lambda ($\lambda$) phosphatase (NEB) for 24 hr at 4°C. The assay was halted by adding 5 µl of 6× Laemmli buffer with 10 mM DTT. Samples were boiled and 10 µl of sample was then used in SDS–PAGE.

## DNA transfections and infections for microscopy

HeLa cells were seeded onto glass coverslips and transfected with YFP-CK2 (α or β), CFP-VAP, YFP-VAP, or YFP-VAP$_{K50L}$ the following day. Twenty-four hours post-transfection, cells were infected with the indicated strain of *C. trachomatis* at a MOI of 1. Twenty-four hours postinfection, media containing 20 ng/ml aTc (final concentration) was added for 4 hr to induce expression of IncV-3xFLAG.

## Immunofluorescence and confocal microscopy

HeLa cells seeded on glass coverslips and infected with *C. trachomatis* were fixed 24 hr postinfection with 4% paraformaldehyde in 1× PBS for 20 min at room temperature then washed with 1× PBS three times. The coverslips were sequentially incubated with primary and secondary antibodies in 0.1% Triton X-100 in 1× PBS for 1 hr at room temperature. For coverslips stained with anti-CK2β, antibodies were diluted in 0.5% Triton X-100% and 5% bovine serum albumin (BSA) in 1× PBS. Coverslips were washed with 1× PBS three times then mounted with glycerol containing DABCO and Tris pH 8.0. Confocal images were obtained using an Andor iXon ULTRA 888BV EMCCD camera and a Yokogawa CSU-W1 Confocal Scanner Unit attached to a Leica DMi8 microscope. 1 μm thick Z slices covering the entirety of the cell were captured. Image analysis was performed using the Imaris software. All the micrographs within a given figure panels are at the same scale.

## Quantification of YFP-CK2β, CFP-VAP, and YFP-VAP inclusion association

Quantification was performed using the Imaris imaging software. A step-by-step illustration of the quantification method is presented in *Figure 3—figure supplement 2*. Quantification of the IncV-3xFLAG associated with the inclusion was performed to ensure there was no defect in inclusion localization (*Figure 3—figure supplement 2A*). The sum of the voxels corresponding to the IncV-3xFLAG signal above the threshold set by the signal within the cytosol was calculated for IncV-3xFLAG and mCherry. Objects were edited such that IncV-3xFLAG colocalizing with the mCherry bacteria was removed. The IncV-3xFLAG volume was normalized to its corresponding inclusion volume to determine the inclusion association of IncV in arbitrary units [au]. The association of a given marker was then determined as follow (*Figure 3—figure supplement 2B*, YFP-VAP is shown as an example). Within the three-dimensional (3D) object generated from the raw signal of IncV-3xFLAG at the inclusion membrane, the mean intensity of the marker of interest was calculated and normalized to the mean intensity of the marker of interest within the cytosol surrounding the inclusion (average of 10 small 3D spheres) to determine the inclusion association of the marker of interest in au. A similar approach is used in *Figure 4E*, except that the mean intensity of CFP-VAP was determined within 3D objects generated from the raw signal of the YFP-CK2 at the inclusion and normalized to the CFP-VAP mean intensity in the cytosol.

Each experiment was performed in triplicate with at least 20–30 inclusions analyzed per condition per replicate. Unless specified, data from three independent replicates are combined into a single graph. Each point on the graph represents a single inclusion with the average value and SEM shown. Student's *t*-tests or one-way ANOVA with multiple comparisons was performed.

## Protein purification

Expression of GST, GST-VAP$_{MSP}$, GST-IncV$_{167-363}$, GST-IncV$_{167-363}$ S/D, or MBP-VAP$_{MSP}$ was induced for 2 hr by the addition of isopropyl-β-ᴅ-thiogalactopyranoside (0.1 mM, final concentration) to a 10 ml culture of *E. coli* BL21-λDE3 at OD 0.8. Bacterial pellets were stored at −80°C. Frozen pellets were thawed and resuspended in 800 μl sonication buffer (20 mM Tris pH 7.5, 300 mM NaCl, 2 mM EDTA, 1 mM MgCl$_2$, 1% Triton X-100, 1 mM DTT, 1 mM PMSF (phenylmethylsulfonyl fluoride)). The samples were sonicated using five 5-s pulses at 40% power then centrifuged at 13,000 × *g* for 10 min at 4°C. 40 μ of glutathione Sepharose beads (GE) for GST-tagged constructs and 40 μl of Amylose resin for MBP-tagged constructs were washed three times with sonication buffer then added to the cleared lysate and incubated for 2 hr at 4°C with rotation. The beads were washed three times in TBS.

## In vitro kinase assay

Protein-bound glutathione Sepharose beads were resuspended in 1× NEBuffer for Protein Kinases supplemented with 1 mM ATPγS and 10 units of CK2 (NEB) and incubated at 30°C for 45 min. P-nitrobenzyl mesylate (PNBM) was added to the kinase reaction at a final concentration of 2.4 mM for 2 hr at room temperature in the dark. The PNBM alkylation reaction was quenched by adding an equal volume of 2× Laemmli buffer. Proteins were separated using SDS–PAGE on a 12% acrylamide gel then transferred to a nitrocellulose membrane. The membrane was stained with Ponceau S in 5% acetic acid to detect total protein then washed in dH$_2$O. The membrane was then probed with

anti-thiophosphate ester antibodies to detect phosphorylated proteins which were detected with HRP-conjugated secondary antibodies.

## In vitro binding assay

First, GST, GST-IncV$_{167-363}$, GST-IncV$_{167-363}$ S/D, and MBP-VAP$_{MSP}$ were purified as described in protein purification. MBP-VAP$_{MSP}$ was eluted from amylose resin using 100 µl 1× TBS supplemented with 10 mM maltose monohydrate. GST, GST-IncV$_{167-363}$, or GST-IncV$_{167-363}$ S/D attached to glutathione beads were washed three times in sonication buffer. 500 µl of sonication buffer containing 1.25 µg MBP-VAP$_{MSP}$ was added to each tube with GST beads and binding was allowed to occur overnight at 4°C with rotation. Following overnight binding, beads were washed three times in 1× TBS. After the final wash, all liquid was removed from the beads which were then suspended in 20 µl 2× Laemmli buffer. The entire sample was separated by SDS–PAGE, proteins transferred to a nitrocellulose membrane which was stained with Ponceau S to detect the GST construct then probed with anti-MBP to detect MBP-VAP$_{MSP}$.

## In vitro binding assay with IncV dephosphorylation

First, the phosphatase assay was performed with the following changes: 1,000,000 HEK293 cells stably transfected with pCMV-IE-N2-IncV-3xFLAG were seeded per 6 well. Six wells were lysed in 500 µl lysis buffer each and lysates from two wells were combined. 10 µl of anti-FLAG beads were added per 1000 µl cleared lysate for 2 hr at 4°C with rotation. All beads were combined after the first wash, and proteins were eluted in 150 µl elution buffer (130 µl eluate collected).

Next, GST and GST-VAP$_{MSP}$ were purified as described in protein purification. Per phosphatase assay tube: 1.5 µg of GST or GST-VAP$_{MSP}$ attached to beads (determined empirically by comparison of Coomassie stained gel to BSA standard curve) was suspended in 500 µl lysis buffer then added to tubes containing the IncV-3xFLAG-containing eluate (±phosphatase treatment). Binding was allowed to occur overnight at 4°C with rotation.

To confirm that IncV dephosphorylation was successful, a set of control tubes were incubated with beads alone (no GST construct) in lysis buffer to mimic experimental conditions.

Twenty-four hours after binding, beads were washed three times in 1× TBS. After the final wash, all liquid was removed from the beads which were then suspended in 20 µl 2× Laemmli buffer. The entire sample was separated by SDS–PAGE, proteins transferred to a nitrocellulose membrane which was stained with Ponceau S to detect the GST construct then probed with anti-FLAG to detect IncV-3xFLAG.

## In vitro binding assay with CK2 phosphorylation of IncV

First, GST, GST-IncV$_{167-363}$, and MBP-VAP$_{MSP}$ were purified as described in protein purification. MBP-VAP$_{MSP}$ was eluted from amylose resin using 100 µl 1× TBS supplemented with 10 mM maltose monohydrate. 1.5 µg of GST or GST-IncV$_{167-363}$ attached to glutathione beads (determined empirically by comparison of Coomassie stained gel to BSA standard curve) or beads alone was suspended in 1× NEBuffer for Protein Kinases with 200 µM ATP (Thermo) and 100 units of CK2 (NEB) at 30°C for 45 min. Beads were washed three times in sonication buffer. 1.25 µg MBP-VAP$_{MSP}$ suspended in 500 µl sonication buffer was added to each tube with beads and binding was allowed to occur overnight at 4°C with rotation. Twenty-four hours after binding, beads were washed three times in 1× TBS. After the final wash, all liquid was removed from the beads which were then suspended in 20 µl 2× Laemmli buffer. The entire sample was separated by SDS–PAGE, proteins transferred to a nitrocellulose membrane which was stained with Ponceau S to detect the GST construct then probed with anti-MBP to detect MBP-VAP$_{MSP}$.

## GST-pull-down immunoblot quantification

Immunoblots and Ponceau S staining were quantified using the ImageJ software (NIH). The immunoblot band intensity was normalized to the Ponceau S band intensity and the fold change determined relative to wild-type or untreated conditions.

## CK2 depletion using siRNA

CK2 was depleted from cells using a pool of four siRNA duplexes or each duplex individually that was transfected with Dhamafect 1 transfection reagents. On day 0, one volume of 200 nM siRNA

in siRNA buffer was incubated with one volume of 5 µl/ml of Dharmafect 1 transfection reagent in DMEM high glucose in a well for 20 min at room temperature. Two volumes of DMEM High Glucose supplemented with 20% fetal bovine serum and 200,000 HeLa cells per ml were added to the well. Cells were incubated at 37°C with 5% $CO_2$ for 3 days. The total volume for one 96 well was 120 µl. The *CSNK2B* target sequence for each individual siRNA duplex was: A, CAACCAGAGUGACCUGAUU; B, GACAAGCUCUAGACAUGAU; C, CAGCCGAGAUGCUUUAUGG; D, GCUCUACGGUUUCAAGAUC. The efficacy of the knock down was quantified using the ImageJ software (NIH). The CK2 band intensity was normalized to the GAPDH band intensity and the knockdown efficacy was determined relative to the mock condition.

## CK2 inhibition using CX-4945 or GO289

CK2 was inactivated using the CK2-specific inhibitor CX-4945 (0308, Advanced Chemblocks) or GO289 (17586, AOBIOUS), as described in the result sections.

## Electron microscopy

HeLa cells were infected with a strain of *C. trachomatis* expressing IncV$_{WT}$ or IncV$_{S3A}$-3xFLAG under the control of the aTc-inducible promoter. Twenty hours postinfection cells were incubated in the presence of 20 ng/ml of aTc. Twenty-four hours postinfection, the samples were fixed in 2.5% glutaraldehyde in 0.1 M cacodylate buffer (pH 7.4) for 60 min at RT. After fixation, cells were scraped off and pelleted, as described previously (*Fölsch et al., 2001*). Ultrathin sections of infected cells were stained with osmium tetraoxide before examination with Hitachi 7600 EM under 80 kV equipped with a dual AMT CCD camera system. Quantitative measurement of length for the inclusion membrane and host ER elements attached to this membrane was performed using ImageJ on 24 representative electron micrographs at low magnification to ensure the entire inclusion fit into the field of view.

## Acknowledgements

We thank Ted Hackstadt (NIH-Rocky Mountain Laboratories) and Mary Weber (University of Iowa) for the *incV::bla* strain; Claude Cochet and Odile Filhol-Cochet (Institut Albert Bonniot Département Réponse et Dynamique Cellulaires) for the YFP-CK2 constructs; David Brautigan (University of Virginia) for the CX-4945 inhibitor; the excellent technical staff of the Electron Microscopy Core Facility at the Johns Hopkins University School of Medicine Microscopy Facility; and Hervé Agaisse and members of the Derré and Agaisse laboratories (University of Virginia) for providing feedback on the project and the manuscript.

## Additional information

### Funding

| Funder | Grant reference number | Author |
| --- | --- | --- |
| National Institute of Allergy and Infectious Diseases | AI007046 | Rachel J Ende<br>Rebecca L Murray<br>Samantha K D'Spain |
| National Institute of Allergy and Infectious Diseases | AI136283 | Rebecca L Murray |
| National Institute of Allergy and Infectious Diseases | AI060767 | Isabelle Coppens |
| National Institute of Allergy and Infectious Diseases | AI101441 | Isabelle Derré |
| National Institute of Allergy and Infectious Diseases | AI162758 | Isabelle Derré |
| National Institute of Allergy and Infectious Diseases | AI141841 | Isabelle Derré |

| Funder | Grant reference number | Author |
| --- | --- | --- |
| National Institute of Allergy and Infectious Diseases | AI166237 | Isabelle Derré |

The funders had no role in study design, data collection, and interpretation, or the decision to submit the work for publication.

## Author contributions

Rachel J Ende, Conceptualization, Formal analysis, Investigation, Methodology, Validation, Visualization, Writing – original draft, Writing – review and editing; Rebecca L Murray, Conceptualization, Formal analysis, Investigation, Methodology, Validation, Visualization, Writing – original draft; Samantha K D'Spain, Formal analysis, Investigation, Validation, Visualization; Isabelle Coppens, Data curation, Formal analysis, Funding acquisition, Investigation, Visualization, Writing – review and editing; Isabelle Derré, Conceptualization, Formal analysis, Funding acquisition, Methodology, Project administration, Supervision, Validation, Visualization, Writing – review and editing

## Author ORCIDs

Rachel J Ende ⓘ http://orcid.org/0000-0002-1625-5433
Isabelle Derré ⓘ http://orcid.org/0000-0002-7055-4346

## Ethics

Ethics statement.All genetic manipulations and containment work were approved by the UVA Biosafety Committee and are in compliance with the section III-D-1-a of the National Institutes of Health guidelines for research involving recombinant DNA molecules.

## Decision letter and Author response

Decision letter https://doi.org/10.7554/eLife.74535.sa1
Author response https://doi.org/10.7554/eLife.74535.sa2

# Additional files

## Supplementary files

- Transparent reporting form
- Supplementary file 1. Primers and cloning strategies.

## Data availability

All data generated or analysed during this study are included in the manuscript and supporting file; Source Data files have been provided for Figures 1, 2, 3, 4, 5, 6, 7, 3S1, 3S3, 4S1, 5S1, 5S2, 5S4, 6S1, 6S2, 6S3, 7S1, 7S2.

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
