## [Editor Report]

This paper describes an interesting case of molecular mimicry where a *Chlamydia* protein needs to be phosphorylated by host kinases in order to interact with an host factor and recruit the host endoplasmic reticulum membrane around bacterial inclusions in the cell. Post-translational modifications of translocated bacterial effectors by host factors have rarely been reported. As such, the story is of interest to those studying microbial pathogenesis and also to cell biologists studying related mechanisms. One limitation of the study is that it relies on overexpression of tagged bacterial proteins. It will therefore be important to confirm these findings in the future with endogenous proteins, when adequate tools become available.

---

## [Decision Letter]

**Decision letter after peer review:**

Thank you for submitting your article "Phospho-regulation accommodates Type III secretion and assembly of a tether of ER-*Chlamydia* inclusion membrane contact sites" for consideration by *eLife*. Your article has been reviewed by 2 peer reviewers, one of whom is a member of our Board of Reviewing Editors, and the evaluation has been overseen by Dominique Soldati-Favre as the Senior Editor. The following individual involved in review of your submission has agreed to reveal their identity: Lena Pernas (Reviewer #2).

Essential revisions:

Although they find the work interesting, the reviewers raise caveats that could invalidate the interpretations of some results. Additional experiments are needed to clarify these. In particular, while the title and abstract of the paper put a great emphasis on membrane contact sites, these sites are, per se, not shown anywhere in the paper, and the effects of IncV phosphorylation on their establishment are inferred indirectly. Another thing requiring clarification is an incongruity with the proposed model; IncV phosphorylated by CK2 in vitro appears to bind VAP efficiently (Figure 3D) despite the fact that CK2 is unlikely to phosphorylate the Threonine residue of the phosphoFFAT, and therefore, that this residue is likely unphosphorylated in this assay. There is therefore a discrepancy with the results of Figure 1, indicating that phosphorylation of the phosphoFFAT is necessary for VAP binding. The same can be said for the phosphomimetic mutant of figure 6D: VAP appears to bind robustly despite the threonine not being phosphorylated.

Here are important points to address:

1) The authors state in the title that "Phospho-regulation accommodates Type III secretion and assembly of a tether of ER-Chlamydia inclusion contact sites." However, nowhere in the manuscript do they actually analyze contact sites (only VAP recruitment by IF). As done in a previous study published by the authors (Stanhope et al., 2017), electron microscopy (EM) in this study is required to depict the lack of ER-contact sites to the *C. trachomatis* inclusion in the absence of IncV phosphorylation.

2) The discrepancy pointed above likely stems from the fact that different assays are used. How would a Threonine to alanine mutant bind VAP in the GST pull down assay? What about a phosphomimetic (see below)?

3) As it stands, the only data supporting a phosphoFFAT are based on a single threonine to alanine mutation. This is insufficient to clearly demonstrate that this is a phosphoFFAT, as a T to A mutation could mess up the protein in various ways, unrelated to phosphorylation. It is therefore important to better characterize the phosphoFFAT motif both in vitro and in vivo. Can a phosphomimetic be generated? phosphomimetic FFAT requires, in addition to mutate the T to D, to mutate the prolines to alanines to allow flexibility in the chain and allow the aspartate to work as a phosphomimic (as explained in di Mattia et al. 2020). Can this rescue VAP recruitment? Alternatively, can the phosphorylation be demonstrated by phosphoproteomics in vivo?

4) The model implies that Phosphorylation of the C-terminal S residues is necessary to trigger the phosphorylation of the phsophoFFAT T-residue. There is however no evidence for that. It is equally plausible that S-residue and T-residue phosphorylation contribute independently to strengthen the interaction with VAP (as suggested by the fact that S-residue phosphorylation promotes VAP interaction independent of T phosphorylation, see Figure 2 and my comment on the discrepancy above). Can phosphoproteomics be used to assess if S-residue mutation or CK2 inhibition affects T phosphorylation? Can a phosphoFFAT phosphomimetic (see above) bypass the need for CK2 phosphorylation? Otherwise, it is equally plausible that the phosphoFFAT is constitutively phosphorylated, and that phosphorylation of the C-terminal S-residues are only necessary to phosphorlylate the S-track around the FFAT motifs. If both models cannot be distinguished, then both should be presented.

5) Using CK2 inhibitors, siRNA, and IncV mutants, the authors arrive at a model where CK2 is recruited to IncV at inclusions (Figure 7, Step 1), phosphorylates IncV (Figure 7 Step 2), and this then enables an interaction between IncV and VAP (Figure 7 Step 3) causing the creation of an MCS. Demonstrating this order of event in vivo as well would strengthen the model. Could the authors provide a time-lapse, or other sort of kinetic analysis of CK2 and VAP recruitment to the bacterial vacuole to support their model that CK2 phosphorylation enables ER tethering (or at least to show that VAP is not recruited prior to CK2)?

6) Related to the point above, the association of CK2 to the inclusion in the presence of CX4945 should be analysed qualitatively and quantitively. The model predicts that CK2 should colocalize with IncV at inclusions whether or not it is active. IncV should bind its recognition sequences on IncV, and unless there is a positive feedback loop (where more phosphorylation causes more recruitment of the kinase), you should be able to observe CK2 at inclusions in the absence of VAP recruitment, showing that subsequent association of VAP to the inclusion depends on CK2-dependent phosphorylation of IncV.

7) The siRNA-mediated knockdown of CK2 was incomplete. If CK2 is not critical for host cell survival, a CRISPR/Casß gene knock-out of CK2α and CK2ß might provide the result that the siRNA approach did not achieve-the authors could then simply assess IncV phosphorylation in 293 cells in the absence of infection.

[Editors' note: further revisions were suggested prior to acceptance, as described below.]

Thank you for resubmitting your work entitled "Phospho-regulation accommodates Type III secretion and assembly of a tether of ER-*Chlamydia* inclusion membrane contact sites" for further consideration by *eLife*. Your revised article has been evaluated by Dominique Soldati-Favre (Senior Editor) and a Reviewing Editor.

You can see that a third reviewer has been brought into the process, who was not able to see and comment on your initial work. This reviewer has a number of concerns, especially concerning the fact that all of the study was performed using overexpressed IncV rather than endogenously expressed from Chlamydia. Although it would be unfair to ask you to address the concerns in full at this stage, we would like to encourage you to look into this matter. In particular, as was already pointed out in the first round of review, it is regrettable that phosphoproteomics proved challenging at nailing an important point of the paper. The reviewer suggests, as an alternative, to utilise anti-phosphoserine antibodies that can be used to probe immunoprecipitated endogenous IncV or to decorate endogenous IncV at the MCS in immunofluorescence.

In addition, reviewer #1 has a remaining unaddressed concern about the function of the phosphoFFAT. Your pull-down assays show that IncV phosphorylated in vitro by CK2 binds robustly to VAP, despite a presumed absence of phosphorylation of the phosphoFFAT (Figure 2D). The reviewer proposes alternative interpretations of the data and suggests performing the pull down with phosphomimic and phosphor-null mutants of IncV (phosphorylated in vitro by CK2) to settle these alternatives.

*Reviewer #1 (Recommendations for the authors):*

This revised version brings new and exciting data, consolidating the story.

The effect of a phospho mutant on the establishment of contact sites is convincing. The partial bypass of CK2 by the FFAT phosphomimic is a compelling evidence that the phosphoFFAT is not constitutively phosphorylated, but requires CK2 phosphorylation on other residues to become phosphorylated. The reorganisation of the manuscript, and keeping the phosphoFFAT to the end is also facilitating the reading.

Further clarification below would help the understanding of this complex story.

– Asked how mutations in the phosphoFFAT motif would influence VAP binding in the GST pull down assay, the author states in their response to the reviewers that "IncVWT does not bind to VAP in this assay". This is demonstrably wrong, as shown in Figure 2D, where IncVWT binds robustly to VAP, provided it was phosphorylated by CK2 beforehand. Since CK2 is unlikely to be responsible for phosphoFFAT phosphorylation, and that the phosphoFFAT is likely unphosphorylated in this assay, then it means that phosphoFFAT phosphorylation is not necessary for VAP binding. There could be several explanations for this surprising result.

a. One possibility is that the binding observed here is suboptimal, because only wrought by the canonical FFAT. In this case a phosphomimic FFAT should increase binding, while a phosphonull FFAT should behave quite exactly like the WT (i.e. this assay should be impervious to the F263A and T265A mutations).

b. Another possibility is that, in contradiction to expectations, CK2 does indeed phosphorylate the FFAT motif, in which case a phosphonull mutation in FFAT should abrogate binding (while a phosphomimic should have no effect).

c. A third possibility is that the phosphoFFAT is necessary for something else than the direct binding of VAP. For instance, it could be necessary in vivo for recruitment or activation of CK2, a step that may be bypassable in vitro given the high amount of CK2.

These should be clarified. The same concern applies for Figure 7D. Here the S/D mutations in the serine track adjacent to the FFATs is sufficient for VAP binding even though the phosphoFFAT motif is clearly non-phosphorylated here. While, here, possibility b. is not existing, a. and c. remain to be clarified. Again, a phosphomimic FFAT should increase binding, while a phosphonull FFAT should behave quite exactly like the WT.

– In the same vein, Figure 5 – supplement figure 4 shows that phosphomimics mutants of the three C-terminal serines do not bind VAP. This is in stark contrast with figure 2D, where CK2-mediated phosphorylation leads to robust VAP binding. This indicates either of two things. Either the phosphomimics mutation do not actually mimic phosphorylation, or CK2 phosphorylates more than the three target sites at the C-terminus. The second explanation is likely favoured by the authors, but this is not explicitly stated. What are the additional residues that could be phosphorylated by CK2? There are arguments against the idea the CK2 phosphorylates the phosphoFFAT. Could CK2 phosphorylate the serine track adjacent to the FFATs?

– Experiment with the VapK50L are not entirely conclusive. Indeed, if the mutant was not able to bind to phosphoFFAT, then this mutant should bind to the T265A mutant as much (or rather, as little) as to WT IncV. This is not the case. Therefore, at face value, the VapK50L appears more like a mutant of general reduced activity (for instance less expressed) than a phosphoFFAT-specific mutant. A positive control would be to test ER-recruitement of a FFAT protein (non-phosphoFFAT), which should bind this mutant as well as a wild-type.

– There is a problem in the assembly of fig6. In panel B, A picture that represented a double mutant in the previous version, now represents a single mutant. Moreover, the double mutant shows robust VAP recruitment.

*Reviewer #3 (Recommendations for the authors):*

In this revised manuscript the authors have performed additional experiments to address the fair comments raised by the previous two referees. This predominantly involved electron microscopy to further establish the link between phospho-IncV and the formation of membrane contact sites (MCS) that form between the *C. trachomatis* inclusion and the endoplasmic reticulum (ER) of the host cell. They also analysed a phosphomimietic mutant.

Strengths:

As a new referee, there are some more general comments on the above and also the content of the original manuscript. Firstly, it should be recognised that the authors have performed a considerable amount of work, including the generation of various genetic tools, to study the role of the virulence factor IncV in the formation of ER-inclusion MCS. Understanding mimicry by bacterial effectors in an important topic, of interest to those studying microbial pathogenesis and also to the cell biology community who study related canonical mechanisms. Indeed, it is often the case that such cellular microbiology experiments illuminate new aspects of fundamental cellular mechanisms as is potentially the case here when studying FFAT motifs and their relationship to VAT and ER-MCS / other MCS in uninfected cells.

Post-translational modifications of translocated bacterial effectors have rarely been reported, and in the case of phosphorylation, the archetypal example is the enteropathogenic *E. coli* translocated intimin receptor (Tir). Reporting another transmembrane effector, this time at the intracellular interface, that becomes phosphorylated during host-pathogen interaction, has in itself significant novelty.

That said, there are striking similarities between IncV FFAT and phospho-FFAT motifs recently identified in eukaryotic cells (Di Mattia et al., EMBO J 2020). While this on the one hand enhances the mimicry story, it dents the novelty of phospho-FFAT per se. It seems the kinases involved in the phosphorylation of phospho-FFAT in cells are yet to be resolved.

Weaknesses:

While this reviewer appreciates the authors attention to detail in trying to study the full biochemical pathway leading to IncV phosphorylation, some intricacies of this story are less well developed and the model in its current form remains speculative. There is no doubt that studying Chlamydia is challenging, but from both the bacterial and host angle there are conclusions drawn from combinations of in vitro and cell culture based approaches, often in combination with overexpression of the virulence factor and/or the putative host targets, which compounds interpretation. I expect as a result, some of the findings correlate with the model, while others do not. Rather than presenting all the current data, it might be preferable to streamline the story focusing on the phosphorylation of IncV to establish firm groundwork for future detailed studies of the mechanism.

To this reviewer, the firmest aspects of the work are that:

- IncV is phosphorylated (Thr/Ser)

- IncV can recruit CK2 via C-terminal Ser residues

- IncV phosphyorylation relates to VAP recruitment and the formation of ER-MCS

A major limitation is the lack of information on the post-translational modifications of endogenous IncV. Which residues are phosphorylated during infection when the protein is not overexpressed? Over-expression (expression from a plasmid in *C. trachomatis*) has been previously shown to enhance MCS formation, and mislocalise or over titrate IncV around the inclusion membrane and one wonders whether this affects the phenotypes. Similarly VAP and CK2 are overexpressed as a fusion protein in most of the experiments, and endogenous CK2 visualised only when IncV-FLAG is over-expressed. Although these approaches implicate IncV, VAP and CK2 in a common event, without true endogenous data details of the mechanistic pathway cannot be inferred.

Largely based on the fact that IncV co-precipitates CK2 when expressed alone in eukaryotic cells (previously published interactome data), the authors speculate on the action of CK2 and host kinases responsible for IncV phosphorylation. This seems quite subjective, especially given the promiscuity and constitutive activation of CK2 (>300 cellular substrates). Their in vitro data support a view that CK2 can bind IncV, but not that this is necessarily the relevant kinase responsible for the phosphorylaton of key FFAT residues in IncV during infection. Why would this be a static recruitment? Did the authors perform any broader kinase screens or use additional kinase inhibitors to verify in a less directed manner the relevance of CK2 and/or implicate other host kinase families?

Also relating to CK2, what is the effect of CK2 knockdown on other substrates and cellular processes? How much CK2 is required in the cell to sustain essential functions in apoptosis and cell proliferation for example? Assuming chemical inhibition also inhibits these processes do they have any pleiotropic effects during infection? What is the effect of CX-4945 and GO289 on known CK2 substrates? How were these inhibitors titrated?

Caution should be taken in the interpretation of in vitro kinase assays. Just because CK2 can phosphorylate IncV in vitro does not in itself mean this is the relevant kinase during infection.

What is the effect of mutating IncV T265 during infection? Is there an influence on infectivity or ER-inclusion MCS?

[Editors' note: further revisions were suggested prior to acceptance, as described below.]

Thank you very much for your letter. We would first like to apologise for the complicated path your manuscript had to go through with the addition of a third reviewer at the revision stage. This third reviewer was involved in reviewing your 1st manuscript but could not manage to complete the evaluation for reasons beyond their control. To avoid any further delay, we therefore had to complete the first round of review without the unique expertise of this reviewer. But because of this expertise, we decided to bring them back at the 2nd round, aware that it created an awkward situation. Because of that, we agreed to not push forward most of the 3rd reviewer's concern and to focus only on one, i.e. that all evidence herein, and in your previous manuscript come from over-expression of the bacterial effector and/or the host targets CK2/VAP.

The model that you are putting forward sets a strong precedent and a bold claim, and surely you would want it to reflect the behaviour of endogenous proteins and not an artefact of protein overexpression.

The reviewer would like to see a simple experiment using wild-type Chlamydia in non-manipulated cells demonstrating that these modifications occur. The reviewer suggests to use an IncV antibody to probe WT chlamydia infections, which, I realise, might not be available, or to purify phosphoproteins and probe if endogenous IncV is in the lot. The concern that overexpression can generate artefacts by triggering non-natural interactions and artificially expanding contact sites is a valid one. Irrespective of the unfortunate circumstances that led this concern to come to light belatedly, this is a point that you should be keen to address unless you had good grounds to dismiss it.

Concerning the point of reviewer #1, we beg to differ that phospho-FFAT is a minor part of your study. Phospho-FFAT is mentioned in your abstract, in three main figures, and is central to your model in figure 8. Right now, your data show a major discrepancy with this model.

The motif that you identify does appear to be necessary for VAP recruitment to inclusions, as a T>A mutation decreases VAP fluorescence there (Figure 6C), and this effect appears to be downstream of the C-terminal serine phosphorylation, since a T>D mutation partially rescues the S3A mutation. This is consistent with the idea that this motif could act as a phosphoFFAT, through which VAP binds IncV.

However, there are also discrepant data; this motif does not need to be phosphorylated to allow IncV-VAP binding (figure 2D, and 7D). Therefore, while this motif appears to be doing something, it is unclear that it acts as a phosphoFFAT. Indeed, the S/D mutations appear to bypass the need for T265 phosphorylation (Figure 7d), which, following the same logic as above, would seem to indicate that the serine track phosphorylation is downstream of T265 phosphorylation, and that this motif doesn't act as a phosphoFFAT but instead as a primer for serine tract phosphorylation.

So, an important unaddressed point remains; is it really a phosphoFFAT? Is it fair to call it a phosphoFFAT? Or is this motif doing something entirely different?

Without evidence that this phosphorylated motif binds VAP directly (hence in vitro), it seems premature to call it a phosphoFFAT. And as you concede, the VAPk50l mutant does little to lift the ambiguity.

We would like to renew our apologies for the way the review process has gone and assure you that this has been beyond our control.

Ultimately, authors and editors want to publish solid manuscripts that stand the test of time. Therefore, we would like to invite you to consider the above concerns and address them the best you can. We are open to further discussions if you think we made an incorrect evaluation of your data, or if other circumstances prevent you from doing requested experiments.

---

## [Author Response]

Essential revisions:Although they find the work interesting, the reviewers raise caveats that could invalidate the interpretations of some results. Additional experiments are needed to clarify these. In particular, while the title and abstract of the paper put a great emphasis on membrane contact sites, these sites are, per se, not shown anywhere in the paper, and the effects of IncV phosphorylation on their establishment are inferred indirectly. Another thing requiring clarification is an incongruity with the proposed model; IncV phosphorylated by CK2 in vitro appears to bind VAP efficiently (Figure 3D) despite the fact that CK2 is unlikely to phosphorylate the Threonine residue of the phosphoFFAT, and therefore, that this residue is likely unphosphorylated in this assay. There is therefore a discrepancy with the results of Figure 1, indicating that phosphorylation of the phosphoFFAT is necessary for VAP binding. The same can be said for the phosphomimetic mutant of figure 6D: VAP appears to bind robustly despite the threonine not being phosphorylated.Here are important points to address:1) The authors state in the title that "Phospho-regulation accommodates Type III secretion and assembly of a tether of ER-Chlamydia inclusion contact sites." However, nowhere in the manuscript do they actually analyze contact sites (only VAP recruitment by IF). As done in a previous study published by the authors (Stanhope et al., 2017), electron microscopy (EM) in this study is required to depict the lack of ER-contact sites to the *C. trachomatis* inclusion in the absence of IncV phosphorylation.

To directly demonstrate that the defect in VAP recruitment observed when IncV phosphorylation is prevented correlates with a defect in ER-Inclusion MCS formation, we performed ultra-structural analysis of cells infected with an IncV mutant complemented with IncV_WT_ or with IncV_S3A_, by electron microscopy. IncV_S3A_ was chosen because mutation of the corresponding serine residues prevents CK2 association with the inclusion, IncV phosphorylation and VAP recruitment to the inclusion (Figure 5B-F). IncV_WT_ was used as a positive control based on our previous published data showing robust MCS formation under these conditions (Stanhope et al., 2017). Quantification of the proportion of inclusion membrane associated with the ER confirmed a reduction in ER-Inclusion MCS upon IncV_S3A_ expression, compared to IncV_WT_. We note that some levels of ER-Inclusion MCS is expected with IncV_S3A_ because of redundant tethering mechanism(s) (Stanhope et al., 2017). The corresponding data is presented in Figure 5G-H and Lines 360-375.

2) The discrepancy pointed above likely stems from the fact that different assays are used. How would a Threonine to alanine mutant bind VAP in the GST pull down assay? What about a phosphomimetic (see below)?

We did not test the IncV_T625A_ mutant in the GST pull down assay. IncV_WT_ does not bind to VAP in this assay, so we did not expect that a loss of function of the phospho-FFAT to lead to VAP binding. Please see point (3) for our response regarding the phosphomimetic mutant.

3) As it stands, the only data supporting a phosphoFFAT are based on a single threonine to alanine mutation. This is insufficient to clearly demonstrate that this is a phosphoFFAT, as a T to A mutation could mess up the protein in various ways, unrelated to phosphorylation. It is therefore important to better characterize the phosphoFFAT motif both in vitro and in vivo. Can a phosphomimetic be generated? phosphomimetic FFAT requires, in addition to mutate the T to D, to mutate the prolines to alanines to allow flexibility in the chain and allow the aspartate to work as a phosphomimic (as explained in di Mattia et al. 2020). Can this rescue VAP recruitment? Alternatively, can the phosphorylation be demonstrated by phosphoproteomics in vivo?

We conducted the following experiments to further characterize the phospho-FFAT.

1- We generated a phosphomimetic mutation (IncV_T265D/P266A_) as explained in Di Mattia et al. (Di Mattia et al., 2020) and tested the ability of the phosphoFFAT phosphomimetic mutant to interact with VAP in vitro and during infection. IncV_T265D/P266A_ was not sufficient to rescue VAP binding in vitro (Figure 6 – supplement figure 2, lines 426-432). However, IncV_T265D/P266A_ was as efficient as IncV_WT_ to recruit VAP to the inclusion (Figure 6B-C, lines 403-425).

2- We tested the recruitment of VAP_K50L_ to the inclusion. The K50L mutation specifically prevents the interaction of VAP with phospho-FFAT motifs but not with canonical FFAT motifs (Di Mattia et al., 2020) (Figure 6D-E, lines 433-444). VAP_K50L_ recruitment to IncV_WT_-positive inclusions was intermediate, due to the contribution of IncV canonical FFAT, which was confirmed by the lack of VAP_K50L_ recruitment to IncV_Y287A_-positive inclusions (Y287A inactivates the canonical FFAT).

Collectively, these results are in line with the characterization of the phospho-FFAT of STARD3 and VPS13D (Di Mattia et al., 2020; Guillen-Samander et al., 2021) and support that the non-canonical FFAT of IncV is a phospho-FFAT that contributes to VAP binding. The corresponding data is presented in Figure 6 – supplemental figure 2 and Figure 6B-E (Lines 388-448) and discussed in the Discussion section (Lines 550-571).

Performing phosphoproteomics has been one of our main goals with this project. Unfortunately, preparing samples to analyze IncV phosphorylation status by phosphoproteomics has been technically challenging and unfruitful so far. This was acknowledged on lines 577-580 of the Discussion section.

4) The model implies that Phosphorylation of the C-terminal S residues is necessary to trigger the phosphorylation of the phsophoFFAT T-residue. There is however no evidence for that. It is equally plausible that S-residue and T-residue phosphorylation contribute independently to strengthen the interaction with VAP (as suggested by the fact that S-residue phosphorylation promotes VAP interaction independent of T phosphorylation, see Figure 2 and my comment on the discrepancy above). Can phosphoproteomics be used to assess if S-residue mutation or CK2 inhibition affects T phosphorylation? Can a phosphoFFAT phosphomimetic (see above) bypass the need for CK2 phosphorylation? Otherwise, it is equally plausible that the phosphoFFAT is constitutively phosphorylated, and that phosphorylation of the C-terminal S-residues are only necessary to phosphorlylate the S-track around the FFAT motifs. If both models cannot be distinguished, then both should be presented.

Please see our response above, in point 3, regarding the technical challenges associated with a phophoproteomics approach.

To addressed if a phospho-FFAT phosphomimetic can bypass the need for CK2, we generated a phopho-FFAT phosphomimetic mutant in the background of the S3A mutation (The S3A mutation prevents CK2 mediated phosphorylation of IncV). Our results indicate an intermediate VAP recruitment to IncV_T265D/P266A/S3A_-positive inclusions, which supports an indirect role for CK2 in the phosphorylation of IncV phospho-FFAT, despite the fact that T265 is not a direct CK2 target. The corresponding data is presented in Figure 6F-G (Lines 449-458) and discussed in the Discussion section (Lines 572-577 ).

5) Using CK2 inhibitors, siRNA, and IncV mutants, the authors arrive at a model where CK2 is recruited to IncV at inclusions (Figure 7, Step 1), phosphorylates IncV (Figure 7 Step 2), and this then enables an interaction between IncV and VAP (Figure 7 Step 3) causing the creation of an MCS. Demonstrating this order of event in vivo as well would strengthen the model. Could the authors provide a time-lapse, or other sort of kinetic analysis of CK2 and VAP recruitment to the bacterial vacuole to support their model that CK2 phosphorylation enables ER tethering (or at least to show that VAP is not recruited prior to CK2)?

Analysis of the sequential recruitment of CK2 and VAP to the inclusion, using time-lapse video microscopy, proved to be technically challenging. Instead, we took advantage of the fact that the kinase activity of CK2 is not required for CK2 recruitment to the inclusion (please see point 6 below). In our experimental setup, we first allowed for CK2 association to the inclusion in the presence of the CK2 inhibitor, which prevents VAP recruitment, and then washed the inhibitor away and assayed for VAP recruitment. Under these conditions, VAP association with the inclusion was observed 1 hour after removal of the CK2 inhibitor, supporting the model that CK2 is recruited first and that phosphorylation enables VAP-dependent ER tethering. The corresponding data is presented in Figure 4C-E (Lines 294-314).

6) Related to the point above, the association of CK2 to the inclusion in the presence of CX4945 should be analysed qualitatively and quantitively. The model predicts that CK2 should colocalize with IncV at inclusions whether or not it is active. IncV should bind its recognition sequences on IncV, and unless there is a positive feedback loop (where more phosphorylation causes more recruitment of the kinase), you should be able to observe CK2 at inclusions in the absence of VAP recruitment, showing that subsequent association of VAP to the inclusion depends on CK2-dependent phosphorylation of IncV.

In Figure 4A-B (Lines 284-293), we provide qualitative and quantitative evidence that the kinase activity of CK2 is not required for CK2 association with the inclusion, supporting the model that CK2 recruitment precede VAP and that VAP is only recruited once CK2 phosphorylates IncV The latter is further supported by our answer to point 5 above.

7) The siRNA-mediated knockdown of CK2 was incomplete. If CK2 is not critical for host cell survival, a CRISPR/Casß gene knock-out of CK2α and CK2ß might provide the result that the siRNA approach did not achieve-the authors could then simply assess IncV phosphorylation in 293 cells in the absence of infection.

When considering our options for genetic targeting of CK2, we favored silencing over a knockout, because ck2 is an essential gene (Buchou et al., 2003; Zhang et al., 2004), but also because we did notice that ck2 silencing was affecting the growth of the cells to some extent. To our knowledge, the only CK2^-/-^ cell line reported so far is a CK2^-/-^ C2C12 myotube cell line (Franchin et al., 2018) and the lack of CK2 was associated with phosphoproteomics perturbations that could potentially confound our results. We also note that the myotube nature of the C2C12 cell line would not be compatible with Chlamydia infection.

To strengthen our results and rule out any potential off target effect of the CK2 inhibitor CX-4945, we complemented our approach by using an independent CK2 inhibitor, GO289 (Borgo et al., 2021). As observed with CX-4945, GO289 prevented IncV phosphorylation and VAP recruitment to the inclusion. The corresponding data is presented in Figure 3 – supplemental figure 3 (Lines 273-277).

References

Borgo, C., Cesaro, L., Hirota, T., Kuwata, K., D'Amore, C., Ruppert, T., Blatnik, R., Salvi, M., & Pinna, L. A. (2021, Mar 15). Comparing the efficacy and selectivity of Ck2 inhibitors. A phosphoproteomics approach. Eur J Med Chem, 214, 113217. https://doi.org/10.1016/j.ejmech.2021.113217

Buchou, T., Vernet, M., Blond, O., Jensen, H. H., Pointu, H., Olsen, B. B., Cochet, C., Issinger, O. G., & Boldyreff, B. (2003, Feb). Disruption of the regulatory β subunit of protein kinase CK2 in mice leads to a cell-autonomous defect and early embryonic lethality. Mol Cell Biol, 23(3), 908-915. https://doi.org/10.1128/MCB.23.3.908-915.2003

Di Mattia, T., Martinet, A., Ikhlef, S., McEwen, A. G., Nomine, Y., Wendling, C., Poussin-Courmontagne, P., Voilquin, L., Eberling, P., Ruffenach, F., Cavarelli, J., Slee, J., Levine, T. P., Drin, G., Tomasetto, C., & Alpy, F. (2020, Dec 1). FFAT motif phosphorylation controls formation and lipid transfer function of inter-organelle contacts. EMBO J, 39(23), e104369. https://doi.org/10.15252/embj.2019104369

Franchin, C., Borgo, C., Cesaro, L., Zaramella, S., Vilardell, J., Salvi, M., Arrigoni, G., & Pinna, L. A. (2018, Jun). Re-evaluation of protein kinase CK2 pleiotropy: new insights provided by a phosphoproteomics analysis of CK2 knockout cells. Cell Mol Life Sci, 75(11), 2011-2026. https://doi.org/10.1007/s00018-017-2705-8

Guillen-Samander, A., Leonzino, M., Hanna, M. G., Tang, N., Shen, H., & De Camilli, P. (2021, May 3). VPS13D bridges the ER to mitochondria and peroxisomes via Miro. J Cell Biol, 220(5). https://doi.org/10.1083/jcb.202010004

Stanhope, R., Flora, E., Bayne, C., & Derre, I. (2017, Nov 7). IncV, a FFAT motif-containing Chlamydia protein, tethers the endoplasmic reticulum to the pathogen-containing vacuole. Proc Natl Acad Sci U S A, 114(45), 12039-12044. https://doi.org/10.1073/pnas.1709060114

Zhang, R., Ou, H. Y., & Zhang, C. T. (2004, Jan 1). DEG: a database of essential genes. Nucleic Acids Res, 32(Database issue), D271-272. https://doi.org/10.1093/nar/gkh024

[Editors' note: further revisions were suggested prior to acceptance, as described below.]

You can see that a third reviewer has been brought into the process, who was not able to see and comment on your initial work. This reviewer has a number of concerns, especially concerning the fact that all of the study was performed using overexpressed IncV rather than endogenously expressed from Chlamydia. Although it would be unfair to ask you to address the concerns in full at this stage, we would like to encourage you to look into this matter. In particular, as was already pointed out in the first round of review, it is regrettable that phosphoproteomics proved challenging at nailing an important point of the paper. The reviewer suggests, as an alternative, to utilise anti-phosphoserine antibodies that can be used to probe immunoprecipitated endogenous IncV or to decorate endogenous IncV at the MCS in immunofluorescence.In addition, reviewer #1 has a remaining unaddressed concern about the function of the phosphoFFAT. Your pull-down assays show that IncV phosphorylated in vitro by CK2 binds robustly to VAP, despite a presumed absence of phosphorylation of the phosphoFFAT (Figure 2D). The reviewer proposes alternative interpretations of the data and suggests performing the pull down with phosphomimic and phosphor-null mutants of IncV (phosphorylated in vitro by CK2) to settle these alternatives.Reviewer #1 (Recommendations for the authors):This revised version brings new and exciting data, consolidating the story.The effect of a phospho mutant on the establishment of contact sites is convincing. The partial bypass of CK2 by the FFAT phosphomimic is a compelling evidence that the phosphoFFAT is not constitutively phosphorylated, but requires CK2 phosphorylation on other residues to become phosphorylated. The reorganisation of the manuscript, and keeping the phosphoFFAT to the end is also facilitating the reading.Further clarification below would help the understanding of this complex story.– Asked how mutations in the phosphoFFAT motif would influence VAP binding in the GST pull down assay, the author states in their response to the reviewers that "IncVWT does not bind to VAP in this assay". This is demonstrably wrong, as shown in Figure 2D, where IncVWT binds robustly to VAP, provided it was phosphorylated by CK2 beforehand. Since CK2 is unlikely to be responsible for phosphoFFAT phosphorylation, and that the phosphoFFAT is likely unphosphorylated in this assay, then it means that phosphoFFAT phosphorylation is not necessary for VAP binding. There could be several explanations for this surprising result.a. One possibility is that the binding observed here is suboptimal, because only wrought by the canonical FFAT. In this case a phosphomimic FFAT should increase binding, while a phosphonull FFAT should behave quite exactly like the WT (i.e. this assay should be impervious to the F263A and T265A mutations).b. Another possibility is that, in contradiction to expectations, CK2 does indeed phosphorylate the FFAT motif, in which case a phosphonull mutation in FFAT should abrogate binding (while a phosphomimic should have no effect).c. A third possibility is that the phosphoFFAT is necessary for something else than the direct binding of VAP. For instance, it could be necessary in vivo for recruitment or activation of CK2, a step that may be bypassable in vitro given the high amount of CK2.These should be clarified. The same concern applies for Figure 7D. Here the S/D mutations in the serine track adjacent to the FFATs is sufficient for VAP binding even though the phosphoFFAT motif is clearly non-phosphorylated here. While, here, possibility b. is not existing, a. and c. remain to be clarified. Again, a phosphomimic FFAT should increase binding, while a phosphonull FFAT should behave quite exactly like the WT.

As stated in our letter, the phospho-FFAT constitutes a minor aspect of our revised manuscript. As such, and based on what is discussed below, we do not believe that additional experiments will significantly change the impact and overall message of our study.

In response to the initial review, we added Figure 6, a multi-panel figure dedicated to the phospho-FFAT which includes a significant amount of data to characterize the phospho-FFAT during infection, using established tools to study phospho-FFAT motifs such as loss- and gain-of-function mutations that affect the phospho-FFAT/VAP interface. Our results, for which we have discussed the limitations, are consistent with phosphorylation of the phospho-FFAT motif during infection. Our conclusions were not challenged by the reviewers. However, in light of reviewer 1 comment’s, we agree to further discuss the potential limitations of the results obtained with the VAP_K50L_ mutant (Please see below).

Reviewer 1 noted some incongruity in our model regarding the role of the phospho-FFAT in the IncV-VAP interaction, because, in vitro, using purified proteins, IncV interacts with VAP only when IncV is phosphorylated by CK2, despite the fact that the phospho-FFAT is not a CK2 target and is therefore presumably not phosphorylated in this particular experiment (Figure 2D). Reviewer 1 concluded that this result indicates that “phospho-FFAT phosphorylation is not necessary for VAP binding”, suggested additional experiments to explain this result, and proposed three possible outcomes with the following explanations: (a) suboptimal binding, (b) CK2 phosphorylation of the phospho-FFAT, and (c) phospho-FFAT involved in something else.

We realize that we may have partially misunderstood points 2 and 3 of the original review. More specifically, in point 2 we took “How would a threonine to alanine mutant bind VAP in the GST pull down assay” literally, and did not understand that by “GST pull down assay”, the reviewer most likely meant “GST pull down assay after in vitro phosphorylation by CK2”. We would like to sincerely apologize to reviewer 1 for this misunderstanding and convey that we never intended to dismiss his/her/their comment.

With regards to further revising our manuscript by performing additional experiments to clarify the role of the phospho-FFAT in VAP binding in vitro, regardless of the outcome of these experiments, the conclusions of our studies will remain unchanged and the manuscript will not be substantially improved, especially because the data presented in figure 6BC and 6FG are consistent with phospho-FFAT phosphorylation during infection, which in our opinion is more relevant to the overall model. We do however agree that the discussion will benefit from presenting the three possible explanations proposed by reviewer 1 when discussing the phosphoFFAT in the context of the result of Figure 2D. See lines 532-541 of revised manuscript.

Similarly, discussing the two possibilities to explain the results with the S/D mutation presented in Figure 7D, seems more appropriate than performing experiments at this stage of the review process since the concern was not raised in the initial review, and here again the outcome will not significantly elevate our study. See lines 541-545 of revised manuscript.

We also note that while the model presented in Figure 8 is derived from the data presented in the manuscript, this model is not set in stone and is mostly intended as a framework to guide future studies. The following has been added to the first paragraph of the discussion after describing the model presented in Figure 8: “This model constitutes a framework to guide futures studies to further dissect the complex phosphorylation events leading to the assembly of the IncV-VAP tether. Below we discuss our results…”. See lines 477-479 of revised manuscript.

– In the same vein, Figure 5 – supplement figure 4 shows that phosphomimics mutants of the three C-terminal serines do not bind VAP. This is in stark contrast with figure 2D, where CK2-mediated phosphorylation leads to robust VAP binding. This indicates either of two things. Either the phosphomimics mutation do not actually mimic phosphorylation, or CK2 phosphorylates more than the three target sites at the C-terminus. The second explanation is likely favoured by the authors, but this is not explicitly stated. What are the additional residues that could be phosphorylated by CK2? There are arguments against the idea the CK2 phosphorylates the phosphoFFAT. Could CK2 phosphorylate the serine track adjacent to the FFATs?

In the result section of the revised manuscript, after presenting the data of Figure 5 – supplement figure 4, we stated “These results indicated that, although critical for CK2 recruitment, IncV hyper-phosphorylation status, and assembly of the IncV-VAP tether at the ER-inclusion MCS, phosphorylation of S_345_, S_346_, and S_350_ alone is not sufficient to promote VAP binding in vitro, suggesting that additional IncV phosphorylation sites are required to promote optimal interaction between IncV and VAP.” Based on the reviewer's comment we have revised our statement to “While we cannot exclude that the phosphomimetic mutations failed to mimic phosphorylation, these results indicate that although critical for CK2 recruitment, IncV hyper-phosphorylation status, and assembly of the IncV-VAP tether at the ER-inclusion MCS, phosphorylation of S_345_, S_346_, and S_350_ alone is not sufficient to promote VAP binding in vitro, suggesting that additional IncV phosphorylation sites are required to promote optimal interaction between IncV and VAP.” See lines 333-334 of revised manuscript.

We have also amended the discussion as follow:

“However, while CK2 was required for ER-Inclusion MCS formation during infection, and sufficient for IncV-VAP interaction in vitro, the introduction of phosphomimetic mutations at S_345_, S_346_, and S_350_ was not sufficient to promote the IncV-VAP interaction in vitro. While it is possible that the phosphomimetic mutations failed to mimic phosphorylation, it is more likely that additional phosphorylation sites exist. Kinase-substrate recognition is a complex process that goes beyond the simple recognition of a consensus sequence and can involve docking sites away from the phosphorylation sites (Miller & Turk, 2018). The cytosolic domain of IncV contains a large number of additional potential CK2 recognition sites. Among those, two serine residues, S_253_ and S_283_, located in the serine tracts upstream of the phospho-FFAT and the canonical FFAT respectively, are direct CK2 target. Additionally, serine residues S_257_, S_258_, and S_259_ (serine tract upstream of the phospho-FFAT), and S_278_, S_281_, S_282_, and S_284_ (serine tract upstream of the canonical FFAT) can be phosphorylated upon priming the serine or tyrosine residues at the last position of the CK2 recognition motif (S/T-x-x-D/E/pS/pY). We therefore propose that alanine substitution of S_345_, S_346_, and S_350_ eliminates an essential docking site for subsequent CK2-mediated phosphorylation of distal residues in the cytosolic domain of IncV, including the serine tracts next to the FFAT motifs (see below). Further investigation of the IncV-dependent recruitment of CK2 to ER-Inclusion MCS could offer some insights into kinase targeting to cellular MCS. Moreover, since intracellular pathogens often mimic cellular processes, our study may have identified CK2 as a regulator of cellular MCS.” See lines 488-501 of revised manuscript.

– Experiment with the VapK50L are not entirely conclusive. Indeed, if the mutant was not able to bind to phosphoFFAT, then this mutant should bind to the T265A mutant as much (or rather, as little) as to WT IncV. This is not the case. Therefore, at face value, the VapK50L appears more like a mutant of general reduced activity (for instance less expressed) than a phosphoFFAT-specific mutant. A positive control would be to test ER-recruitement of a FFAT protein (non-phosphoFFAT), which should bind this mutant as well as a wild-type.

The VAP_K50L_ mutant was chosen for its specificity to phospho-FFAT motifs, based on the data by Di Mattia et al. who showed decreased binding to phospho-FFAT, but not to canonical FFAT motifs by performing the positive control suggested above.

In our assay, the VAP_K50L_ mutant was not less expressed than the WT (see Figure 6 – source data 1; tab Figure 6E; YFP-VAP intensity in the cytosol; the average intensity is the same for VAP_WT_ and VAP_K50L and_).

As opposed to what was shown by Di Mattia et al., our data could be consistent with low residual binding of VAP_K50L_ to IncV phosphoFFAT motifs and/or an overall reduced activity towards the canonical FFAT. We have amended the text accordingly. See lines 398-400 (results) and 517-521 (discussion) of revised manuscript.

– There is a problem in the assembly of fig6. In panel B, A picture that represented a double mutant in the previous version, now represents a single mutant. Moreover, the double mutant shows robust VAP recruitment.

On April 21^st^, we realized that panels had been improperly positioned during the revision of the phospho-FFAT figure (Figure 6) and immediately notified the staff, who uploaded a corrected Figure 6. We apologize for this oversight.

Reviewer #3 (Recommendations for the authors):In this revised manuscript the authors have performed additional experiments to address the fair comments raised by the previous two referees. This predominantly involved electron microscopy to further establish the link between phospho-IncV and the formation of membrane contact sites (MCS) that form between the *C. trachomatis* inclusion and the endoplasmic reticulum (ER) of the host cell. They also analysed a phosphomimietic mutant.Strengths:As a new referee, there are some more general comments on the above and also the content of the original manuscript. Firstly, it should be recognised that the authors have performed a considerable amount of work, including the generation of various genetic tools, to study the role of the virulence factor IncV in the formation of ER-inclusion MCS. Understanding mimicry by bacterial effectors in an important topic, of interest to those studying microbial pathogenesis and also to the cell biology community who study related canonical mechanisms. Indeed, it is often the case that such cellular microbiology experiments illuminate new aspects of fundamental cellular mechanisms as is potentially the case here when studying FFAT motifs and their relationship to VAT and ER-MCS / other MCS in uninfected cells.Post-translational modifications of translocated bacterial effectors have rarely been reported, and in the case of phosphorylation, the archetypal example is the enteropathogenic *E. coli* translocated intimin receptor (Tir). Reporting another transmembrane effector, this time at the intracellular interface, that becomes phosphorylated during host-pathogen interaction, has in itself significant novelty.That said, there are striking similarities between IncV FFAT and phospho-FFAT motifs recently identified in eukaryotic cells (Di Mattia et al., EMBO J 2020). While this on the one hand enhances the mimicry story, it dents the novelty of phospho-FFAT per se. It seems the kinases involved in the phosphorylation of phospho-FFAT in cells are yet to be resolved.Weaknesses:While this reviewer appreciates the authors attention to detail in trying to study the full biochemical pathway leading to IncV phosphorylation, some intricacies of this story are less well developed and the model in its current form remains speculative. There is no doubt that studying Chlamydia is challenging, but from both the bacterial and host angle there are conclusions drawn from combinations of in vitro and cell culture based approaches, often in combination with overexpression of the virulence factor and/or the putative host targets, which compounds interpretation. I expect as a result, some of the findings correlate with the model, while others do not. Rather than presenting all the current data, it might be preferable to streamline the story focusing on the phosphorylation of IncV to establish firm groundwork for future detailed studies of the mechanism.To this reviewer, the firmest aspects of the work are that:- IncV is phosphorylated (Thr/Ser)- IncV can recruit CK2 via C-terminal Ser residues- IncV phosphyorylation relates to VAP recruitment and the formation of ER-MCSA major limitation is the lack of information on the post-translational modifications of endogenous IncV. Which residues are phosphorylated during infection when the protein is not overexpressed? Over-expression (expression from a plasmid in *C. trachomatis*) has been previously shown to enhance MCS formation, and mislocalise or over titrate IncV around the inclusion membrane and one wonders whether this affects the phenotypes. Similarly VAP and CK2 are overexpressed as a fusion protein in most of the experiments, and endogenous CK2 visualised only when IncV-FLAG is over-expressed. Although these approaches implicate IncV, VAP and CK2 in a common event, without true endogenous data details of the mechanistic pathway cannot be inferred.Largely based on the fact that IncV co-precipitates CK2 when expressed alone in eukaryotic cells (previously published interactome data), the authors speculate on the action of CK2 and host kinases responsible for IncV phosphorylation. This seems quite subjective, especially given the promiscuity and constitutive activation of CK2 (>300 cellular substrates). Their in vitro data support a view that CK2 can bind IncV, but not that this is necessarily the relevant kinase responsible for the phosphorylaton of key FFAT residues in IncV during infection. Why would this be a static recruitment? Did the authors perform any broader kinase screens or use additional kinase inhibitors to verify in a less directed manner the relevance of CK2 and/or implicate other host kinase families?Also relating to CK2, what is the effect of CK2 knockdown on other substrates and cellular processes? How much CK2 is required in the cell to sustain essential functions in apoptosis and cell proliferation for example? Assuming chemical inhibition also inhibits these processes do they have any pleiotropic effects during infection? What is the effect of CX-4945 and GO289 on known CK2 substrates? How were these inhibitors titrated?Caution should be taken in the interpretation of in vitro kinase assays. Just because CK2 can phosphorylate IncV in vitro does not in itself mean this is the relevant kinase during infection.What is the effect of mutating IncV T265 during infection? Is there an influence on infectivity or ER-inclusion MCS?

As stated in our letter and acknowledged in the decision letter, we believe that it would be unfair to ask us to experimentally address reviewer 3 concerns at this stage.

However, we have acknowledged in the revised manuscript that the studies were performed under IncV overexpression by adding “Although this model was in part inferred from overexpression studies, it constitutes a framework to guide futures studies to further dissect the complex phosphorylation events leading to the assembly of the IncV-VAP tether”. See lines 476-479 of revised manuscript.

[Editors' note: further revisions were suggested prior to acceptance, as described below.]

The model that you are putting forward sets a strong precedent and a bold claim, and surely you would want it to reflect the behaviour of endogenous proteins and not an artefact of protein overexpression.The reviewer would like to see a simple experiment using wild-type Chlamydia in non-manipulated cells demonstrating that these modifications occur. The reviewer suggests to use an IncV antibody to probe WT chlamydia infections, which, I realise, might not be available, or to purify phosphoproteins and probe if endogenous IncV is in the lot. The concern that overexpression can generate artefacts by triggering non-natural interactions and artificially expanding contact sites is a valid one. Irrespective of the unfortunate circumstances that led this concern to come to light belatedly, this is a point that you should be keen to address unless you had good grounds to dismiss it.Concerning the point of reviewer #1, we beg to differ that phospho-FFAT is a minor part of your study. Phospho-FFAT is mentioned in your abstract, in three main figures, and is central to your model in figure 8. Right now, your data show a major discrepancy with this model.The motif that you identify does appear to be necessary for VAP recruitment to inclusions, as a T>A mutation decreases VAP fluorescence there (Figure 6C), and this effect appears to be downstream of the C-terminal serine phosphorylation, since a T>D mutation partially rescues the S3A mutation. This is consistent with the idea that this motif could act as a phosphoFFAT, through which VAP binds IncV.However, there are also discrepant data; this motif does not need to be phosphorylated to allow IncV-VAP binding (figure 2D, and 7D). Therefore, while this motif appears to be doing something, it is unclear that it acts as a phosphoFFAT. Indeed, the S/D mutations appear to bypass the need for T265 phosphorylation (Figure 7d), which, following the same logic as above, would seem to indicate that the serine track phosphorylation is downstream of T265 phosphorylation, and that this motif doesn't act as a phosphoFFAT but instead as a primer for serine tract phosphorylation.So, an important unaddressed point remains; is it really a phosphoFFAT? Is it fair to call it a phosphoFFAT? Or is this motif doing something entirely different?Without evidence that this phosphorylated motif binds VAP directly (hence in vitro), it seems premature to call it a phosphoFFAT. And as you concede, the VAPk50l mutant does little to lift the ambiguity.We would like to renew our apologies for the way the review process has gone and assure you that this has been beyond our control.Ultimately, authors and editors want to publish solid manuscripts that stand the test of time. Therefore, we would like to invite you to consider the above concerns and address them the best you can. We are open to further discussions if you think we made an incorrect evaluation of your data, or if other circumstances prevent you from doing requested experiments.

We have addressed the comments as follow.

– The IncV overexpression conditions is now acknowledged in the discussion (Lines 472-274).

– We have toned down our conclusions regarding the phospho-FFAT. We refer to this motif as the non-canonical FFAT instead of a phospho-FFAT, and discuss the role of T265 in VAP recruitment to the inclusion either as part of a phospho-FFAT or as an independent mechanism.

We have reorganized Figure 6. Because the results were inconclusive, panels 6DE (VAPK50L) have been moved to a new Figure 6 – supplement figure 3; the source data have been edited accordingly. The phospho-FFAT has also been removed from our model in Figure 8 and the related text has been modified accordingly. (Results lines 340-414 and Discussion Lines 505-563)